# Using Saildrones to Validate Satellite-Derived Sea Surface Salinity and Sea Surface Temperature along the California/Baja Coast

**Jorge Vazquez-Cuervo** [1,*]**, Jose Gomez-Valdes** [2] **, Marouan Bouali** [3] **, Luis E. Miranda** [2] **, Tom Van der Stocken** [1] **, Wenqing Tang** [1] **and Chelle Gentemann** [4]

1   Jet Propulsion Laboratory, California Institute of Technology, Pasadena, CA 91109, USA
2   Physical Oceanography Department, Center for Scientific Research and Higher Education at Ensenada, Ensenada 22860, Baja California, Mexico
3   Institute of Oceanography, University of São Paulo, São Paulo 05508-120, Brazil
4   Earth and Space Research, 2101 Fourth Avenue, Suite 1310, Seattle, Washington, WA 98121, USA
*   Correspondence: jorge.vazquez@jpl.nasa.gov; Tel.: +1-818-354-6980

**Abstract:** Traditional ways of validating satellite-derived sea surface temperature (SST) and sea surface salinity (SSS) products by comparing with buoy measurements, do not allow for evaluating the impact of mesoscale-to-submesoscale variability. We present the validation of remotely sensed SST and SSS data against the unmanned surface vehicle (USV)—called Saildrone—measurements from the 60 day 2018 Baja California campaign. More specifically, biases and root mean square differences (RMSDs) were calculated between USV-derived SST and SSS values, and six satellite-derived SST (MUR, OSTIA, CMC, K10, REMSS, and DMI) and three SSS (JPLSMAP, RSS40, RSS70) products. Biases between the USV SST and OSTIA/CMC/DMI were approximately zero, while MUR showed a bias of 0.3 °C. The OSTIA showed the smallest RMSD of 0.39 °C, while DMI had the largest RMSD of 0.5 °C. An RMSD of 0.4 °C between Saildrone SST and the satellite-derived products could be explained by the diurnal and sub-daily variability in USV SST, which currently cannot be resolved by remote sensing measurements. SSS showed fresh biases of 0.1 PSU for JPLSMAP and 0.2 PSU and 0.3 PSU for RMSS40 and RSS70 respectively. SST and SSS showed peaks in coherence at 100 km, most likely associated with the variability of the California Current System.

**Keywords:** MODIS; oceanography; remote sensing; saildrone; sea surface salinity; sea surface temperature; SMAP; validation

## 1. Introduction

As a motivating factor in the study, the application of remote sensing techniques for understanding coastal and open-ocean surface water properties is an area of active research, helping to better understand oceanic variability associated with mesoscale and submesoscale variability such as frontal structures, eddies, and meanders. These features are associated with upwelling and downwelling, and have been recognized to play an important role in shaping physical and biogeochemical processes in the ocean [1] and influencing the spatiotemporal variability in primary productivity levels [2,3]. Typically, these mesoscale and submesoscale features reveal a clear signature in sea surface temperature (SST) [4] and sea surface salinity (SSS) [5]. Hence, continued efforts are needed to characterize and observe these structures and improve the validation quality of remotely sensed SST and SSS observations.

Traditionally, the validation of SST and SSS data has been achieved by direct comparisons with oceanic buoy measurements [6–8]. However, this approach does not allow for determining

how well remote sensing data is resolving the spatial variability at the mesoscale to submesoscale. A reprocessing of the Advanced Very High-Resolution Radiometer (AVHRR) dataset, from 9 km to 4 km, reduced the biases and standard deviations when compared with in situ data from the World Ocean Database (WOD) [9]. For two regions in the California Current System and the Gulf Stream, considerable differences were found in regional SST gradients calculated from the Moderate Resolution Imaging Spectroradiometer (MODIS) and AVHRR data [10]. Similar results were also found in a study off the Peruvian–Chilean coast, a major upwelling region, where the order of magnitude differences in SST gradients were explained by differences in the resolution of the SST datasets [11]. Also, off the Peruvian–Chilean coast, high-resolution numerical ocean models were used for validating remotely sensed SST gradients and revealed that SST gradients are related to changes in the seasonal upwelling cycle [12]. Other studies have shown the importance of using shipborne sensors and other instruments that can resolve the spatial variability to validate satellite-derived SST measurements [13]. Recently, a multi-stage trigonometric interpolation technique was applied to determine the subpixel variability of satellite SST data, illustrating that this methodology could be used to examine striping artifacts in MODIS data, as well as possible features associated with cloud contamination [14]. In the Arctic Ocean, measurements from the Ball Experiment SST (BESST) thermal infrared radiometer were compared against MODIS data and revealed significant spatial variability within 1 km pixels that were associated with density fronts in the marginal ice zone [15]. Hence, methodologies must include understanding the subpixel scale variability, especially in areas of high spatial variability such as in upwelling and downwelling regions. Additionally, SST analyses are likely to perform differently depending on the environmental conditions [6], stressing the need for independent coastal validation.

Validation of SSS products has been a more recent research topic with the launch of the European Space Agency's (ESA) Soil Moisture Ocean Salinity (SMOS) mission, the Aquarius mission, a collaboration between the National Aeronautics and Space Administration (NASA) and the Space Agency of Argentina (Comisión Nacional de Actividades Espaciales, CONAE), and NASA's Soil Moisture Active Passive (SMAP) mission. With the launch of these missions and improvements in calibration, applications to coastal areas have provided opportunities to directly compare satellite-derived SSS with in situ data. For global comparisons, SSS has been validated using ARGO [7,8], while thermosalinographs (TSG) have been extensively used for validation on mesoscales [16]. Inter-comparisons of the SMAP and SMOS data in both the Bay of Bengal and the Gulf of Mexico revealed standard deviation differences of less than 0.3 PSU [16–18]. A primary conclusion of the work was that satellite-derived SSS has the potential to observe freshwater plumes associated with river discharge. These studies [16,17] showed that vertical and horizontal gradients complicated the validation of SSS due to the stratification. On a global scale [16], it was found that biases were dependent on latitude and strongly increased near land masses. Results were based on comparisons with traditional pointwise measurements made by in situ buoys. However, the results are consistent in the finding that biases and standard deviations increased at distances less than 100 km from land.

Overall, the results from these studies point to the need for validation strategies in critical parts of the world's oceans, including Arctic and coastal regions. Additionally, these strategies must go beyond traditional point measurements. Unmanned surface vehicles (USVs), such as Saildrones, provide a novel sampling technique that allows resolving mesoscale and submesoscale processes and, thus, validating the capability of remote sensing data to capture the spatial variability observed in coastal regions. Saildrones provide in situ sampling at high spatial (<1 km) and temporal (1 min) resolution. In this study, we validated satellite-derived SST and SSS measurements in a major coastal upwelling region off the California and Baja coast and, thus, show the capabilities of USVs for such studies. This region was deliberately considered as it is dominated by mesoscale to submesoscale variability associated with the California Current System [19,20], and provides a perfect test case for validating the performance of remote sensing tools to observe SST and SSS in regions with high oceanographic complexity. Although the Saildrone deployment also measures ocean chlorophyll-a concentration, the focus of this work was on SST and SSS, primarily because they both are available as

gridded Level 4 (gapless) products. For SST, multi-sensor optimally interpolated Level 4 products are available through the Group for High-Resolution Sea Surface Temperature (GHRSST), while SSS data are available as Level 3 products, but with few data gaps over an 8 day average. In contrast, chlorophyll-a has significant data gaps in the region due to the persistent cloud cover, requiring more rigorous co-location strategies. Hence, the validation of the chlorophyll-a product is best left as a separate topic and focus. The primary goal here was twofold:

1.　Demonstrate the feasibility of using the Saildrone USV to validate satellite-derived SST and SSS in a coastal upwelling region;
2.　Once validated, examine how well the satellite-derived SST and SSS are representing the spatial variability of the California/Baja Coast.

## 2. Methods and Materials

### 2.1. Remote Sensing and Saildrone Data

#### 2.1.1. Saildrone SSS and SST Data

Saildrone is a remotely controlled wind- and solar-powered USV capable of long-distance deployments lasting up to 12 months and providing high-quality, near real-time, multivariate surface–ocean and atmospheric observations while transiting at typical speeds of ~5 to 9 km/h. Data used in this work were from the California/Baja deployment from 11 April to 11 June 2018. The Saildrone Baja campaign consisted of a 60 day cruise from San Francisco Bay, down along the US/Mexico coast to Guadalupe Island and back. For validation purposes during the Baja deployment, the Saildrone data were compared directly with buoys from the National Data Buoy Center (NDBC). The instrument itself travels at 2 m/s, i.e., 120 m in one minute.

While Saildrones measure a broad set of environmental variables, including air pressure, wind speed, oxygen, chlorophyll-a, humidity, and air temperature, the focus of this work was to validate remotely sensed SSS and SST measurements. Saildrone SSS and SST measurements were both derived from the onboard conductivity, temperature, depth (CTD) sensors, and were measured at 0.6 m depth. Saildrones also carry an onboard Acoustic Doppler Current Profiler (ADCP), which measures horizontal and vertical velocity. The SSS and SST data were used in this study, but the ADCP and other environmental data were not.

The accuracy of Saildrone USV salinity observations was examined using nearby ship observations for the period 1 May to 10 May 2015 [21]. This was for a different time and region than the Baja California campaign, but, nonetheless, should be reflective of the overall accuracy of the Saildrone instrument. The root mean square (RMS) salinity difference was found to be 0.01 PSS-78. A longer two-month validation was completed in 2018, and all sensors, except the skin SST and barometric pressure, were found to be operating within the manufacturer's specifications [22].

This Saildrone Baja dataset was comprised of one data file with the Saildrone platform telemetry and near-surface observational data (air temperature, sea surface skin and bulk temperatures, salinity, oxygen and chlorophyll-a concentrations, barometric pressure, wind speed and direction) for the entire cruise at 1 min temporal resolution. All data files were in netCDF format and CF/ACDD compliant consistent with the NOAA/NCEI specification. For more information and data access, the reader is referred to Reference [23].

#### 2.1.2. Remotely Sensed SSS Data

Three different remote sensing SSS datasets were used in this study: (1) the Jet Propulsion Laboratory version 4.0 SMAP (JPLSMAP) dataset; (2) the Remote Sensing Systems version 3.0 40 km (RSS40) dataset; and (3) the Remote Sensing Systems version 3.0 70 km (RSS70) dataset. Thus, the three SSS datasets were all SMAP derived.

The JPLSMAP product has an inherent spatial resolution of 60 km. All the datasets were gridded at 25 km, with daily files produced as 8 day running means. The rationale for the 8 day averaging is that the SMAP satellite has an inherent repeat orbit of 8 days. For more information on the JPLSMAP product the reader is referred to Reference [24].

The RSS40 and RSS70 datasets have an inherent spatial resolution of 40 km and 70 km, respectively. The rationale for the two products was to allow for retrievals closer to land. Land contamination is the leakage of energy from land surface into the radiometer receiver through side lobes or partially through the main lobe of the SMAP radiometer. In fact, SMOS suffers more from land contamination than SMAP and as far as 1000 km from land, due to the interferometric measurement method. Warmer land temperatures mixed with ocean signature causes large errors in salinity retrieval near land. The so-called land correction is a step in the retrieval algorithm to remove the land contribution from the radiometer measured brightness temperature (BT) before it is used for ocean salinity retrieval. Teams at JPL and RSS developed the land correction scheme using different approaches. Basically, JPL's land correction is derived from SMAP data itself [25], while RSS' land correction is based on simulated land brightness temperature [24].

The 40 km product allowed for retrievals closer to land, but with less smoothing than the lower resolution 70 km product. For specific information on the 40 km and the 70 km product, the reader is referred to References [26–28], respectively. User guides, as well as the "Algorithm Theoretical Basis Document" (ATBD) for these datasets, may be found under the docs directory [29].

### 2.1.3. Remotely Sensed SST Data

Six different Level 4 SST products from the Group for High-Resolution Sea Surface Temperature (GHRSST) were used in this study, the Multi-Scale Ultra-High-Resolution (MUR) Sea Surface Temperature, the Operational Sea Surface Temperature and Sea Ice Analysis (OSTIA), the Canadian Meteorological Center (CMC) Sea Surface Temperature dataset, the Remote Sensing Systems microwave/infrared merged sea surface temperature product (REMSS), the Danish Meteorological Institute's (DMI) sea surface temperature product, and the Naval Oceanographic Office (NAVO) K10 product.

The MUR SST data are a Level 4 products that use wavelets in an optimal interpolation approach. Data from the AVHRR, the MODIS, the NASA Advanced Microwave Scanning Radiometer on EOS (AMSR-E), and the US Navy's Windsat are used in the processing of the Level 4 product. For more details on the product, the reader is referred to Reference [30]. A detailed analysis of the algorithm and product validation may be found in Reference [31].

The OSTIA SST is gridded at 5 km and is produced by the UK Meteorological Office. The different sensors used include AVHRR, the Spinning Enhanced Visible and Infrared Imager (SEVIRI), the Geostationary Operational Environmental Satellite (GOES) imager, the Infrared Atmospheric Sounding Interferometer (IASI), the Tropical Rainfall Measuring Mission Microwave Imager (TMI), and in situ data from ships, and drifting and moored buoys. For more information on the OSTIA dataset, the reader is referred to Reference [32]. More information on the processing and algorithm may be found in Reference [33].

The CMC SST data are provided by the Canadian Meteorological Office. Two versions have been produced at 20 km and 10 km, respectively. Currently, in the forward stream, only the 10 km gridded data are produced. That was the version used in this study. Sensors used in the Level 4 product include the AVHRR from NOAA-18 and 19, the European Meteorological Operational-A (METOP-A) and Operational-B (METOP-B), and microwave data from the Advanced Microwave Scanning Radiometer 2 (AMSR2) onboard the GCOM-W satellite in conjunction with in situ observations of SST from drifting buoys and ships from the ICOADS program. For more information and data access, the reader is referred to Reference [34]. Additional information on processing and validation may be found in Reference [35].

The REMSS dataset is produced at Remote Sensing Systems. The product merges both microwave and infrared data using optimal interpolation. Additionally, the product applies a diurnal model to account for day–night differences. Sensors used include: the microwave Global Precipitation Measurement (GPM) Microwave Imager (GMI), the Tropical Rainfall Measuring Mission (TRMM) Microwave Imager (TMI), the NASA Advanced Microwave Scanning Radiometer-EOS (AMSRE), the Advanced Microwave Scanning Radiometer 2 (AMSR2) onboard the GCOM-W1 satellite, WindSat on board the Coriolis satellite, infrared (IR) sensors such as the MODIS on the NASA Aqua and Terra platforms, and the Visible Infrared Imaging Radiometer Suite (VIIRS) on board the Suomi-NPP satellite. For more information and data access, the reader is referred to Reference [36].

The NAVO K10 dataset is produced by the Naval Oceanographic Office (NAVOCEANO). Microwave and infrared sensors are combined in an optimal interpolation. The following sensors are used: the AVHRR, the Advanced Microwave Scanning Radiometer for EOS (AMSR-E), and the Geostationary Operational Environmental Satellite (GOES) Imager. The analysis was tuned to represent SST at one meter. For more information and data access, the reader is referred to Reference [37].

The DMI dataset is produced by the Danish Meteorological Institute. The sensors incorporated in the analysis include the AVHRR, the Spinning Enhanced Visible and Infrared Imager (SEVIRI), the Advanced Microwave Scanning Radiometer 2 (AMSR2), the Visible Infrared Imager Radiometer (VIIRS), and the MODIS on Aqua. For more information and data access, the reader is referred to Reference [38]. More information on this product may also be found in Reference [39].

*2.2. Comparisons between Satellite and Saildrone Data*

The methodological approach considered in this work allowed us to examine the feasibility of using Saildrone data for evaluating the validation quality of satellite-derived SST and SSS data in coastal regions with high oceanographic complexity such as frontal activity.

For both SST and SSS satellite products, the co-location with the Saildrone data was conducted by averaging the values of all pixels located within a spatial box defined by the distance in kilometers, Δd, from the Saildrone measurement for the same day as the satellite map. The Δd was selected as the spatial resolution of a given satellite product. The spatial co-location window was 1 km for MUR, 5 km for OSTIA, 10 km for CMC, 60 km for JPLSMAP SSS, 70 km for RSS70, and 40 km for RSS40. It should be mentioned that co-location in time was restricted to same day measurements, regardless of the smoothing used in the Level 3 (SSS) or Level 4 (SST) product. In fact, SMAP SSS products were produced daily but consisted of 8 day running means. These co-location criteria were applied throughout the study, ensuring that more than one pixel was used for the comparison, thus, reducing any potential impact of noise in the satellite dataset. Co-locations were done for each Saildrone sample to not degrade the resolution of the Saildrone sampling. It is important to note that because the co-location was done for each Saildrone sample, matchups were not necessarily independent due to the much lower resolution of the satellite data sets.

Comparisons were done directly between the satellite-derived products and the Saildrone data. Biases, RMS differences, and overall signal-to-noise ratios were derived for each of the products. The bias was simply defined as:

$$BIAS = \frac{1}{N}\sum_{1}^{N}(SAT - SDRO) \tag{1}$$

where SAT is equal to one of the six parameters (i.e., JPLSMAP, RSS40, RSS70, MUR, OSTIA, and CMC), SDRO is the Saildrone SST or SSS, and $N$ is the total number of matchups based on co-location criteria, between the SDRO and SAT along the Saildrone track. The RMS differences were then calculated as follows:

$$RMSD = \sqrt{\frac{1}{N}\sum_{1}^{N}(SAT - SDRO - BIAS)^2} \tag{2}$$

where BIAS is the satellite–Saildrone bias defined in Equation (1). To evaluate potential noise, when compared to Saildrone as the reference, the signal-to-noise ratio for each satellite parameter was derived as:

$$\frac{S}{N} = \frac{\sqrt{\frac{1}{N}\Sigma_1^N (SDRO - SDROMEAN)^2}}{RMSD} \tag{3}$$

where $\frac{S}{N}$ is the signal-to-noise ratio, and RMSD is the root mean square difference defined in Equation (2). Thus, as defined by Equation (3), SDROMEAN is the mean of the SSS and SST measurements, respectively, over the entire Saildrone deployment, and thus the signal-to-noise ratio is representative over the entire spatial scale.

Spectra and coherences were calculated for the different parameters to determine the relationship between the Saildrone and satellite data at different spatial scales. The parameters included the Saildrone SST and SSS data, RSS40 SSS, RSS70 SSS, JPLSMAP SSS, MUR SST, OSTIA SST, and CMC SST. Examining spectral slope is critical in coastal oceans where spatial scales can vary on the order of meters to hundreds of kilometers.

A major question to be answered was how the observed differences between the satellite-derived and Saildrone products related to unresolved subpixel scale spatial variability. To further examine the differences between the satellite-derived products and Saildrone measurements, and their relationship to spatial scales, the wavelength spectra were examined for consistency with known spectral slopes defining mesoscale–submesoscale variability. Overall, the goal was to connect these differences to possible issues of spatial resolution inherent to each dataset.

A first simple test was to compare the daily variability of the Saildrone data with the RMSD between the satellite-derived products and Saildrone. One question to be addressed was whether the inherent daily variability of the Saildrone-derived SST and SSS could explain the discrepancies among the products.

## 3. Results

### 3.1. Spatial and Temporal Representation

The SST patterns from Saildrone, MUR, OSTIA, and CMC showed a strong similarity with warmer waters off Southern California and the Baja coasts (Figure 1). All the SST datasets showed similar results. Thus, only MUR, OSTIA, and CMC are shown as examples of the SST results. All products clearly showed cooler waters along the coasts north of 32° N. These waters are associated with the coastal seasonal upwelling and are consistent with the known seasonal cycle [20,40]. All the products also showed the strong gradient in SST south of 32° N. The CMC product showed slightly cooler SSTs along the westward track at 36° N. Statistical comparisons between these products are shown in Section 3.2 Statistical Comparisons.

Compared to discrepancies found between the SST satellite and Saildrone products, larger discrepancies were found between SSS measurements from the Saildrone, the JPLSMAP, RSS40, and the RSS70 product (Figure 2). The RSS40 and RSS70 products clearly show fresher biases near the coast. Even before statistics are calculated among the products, one can visually observe a strong similarity between the Saildrone CTD SST and the satellite SST products, while the same imagery of the SSS indicates that pronounced biases exist among the products. Overall, one identifies visually the fresh bias between the RSS40, RSS70, and the Saildrone SSS. The JPLSMAP product showed a salty signal near the coast at 35° N, which did not appear in the Saildrone SSS measurements. The RSS40 product showed a salty signal near 32° N (see Figure 2) and near the coast which did not appear in the Saildrone SSS data. To better identify the temporal evolution of the signal, the time-series of the products over the deployment period were examined.

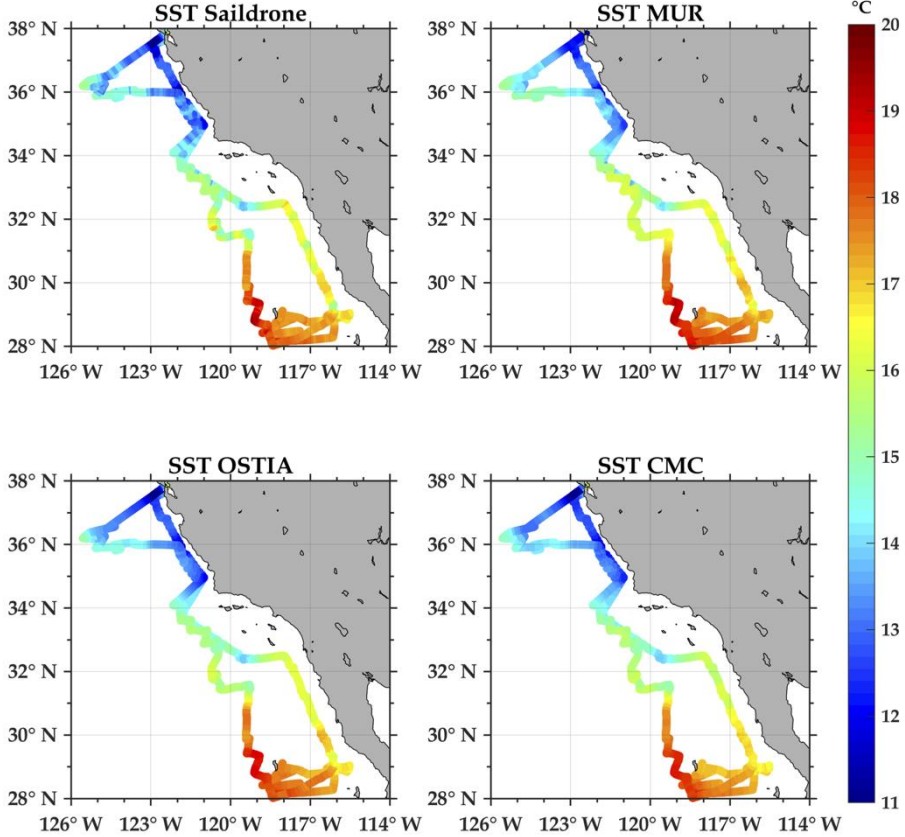

**Figure 1.** Sea surface temperature (SST) values from a Saildrone CTD, MUR, OSTIA, and the CMC.

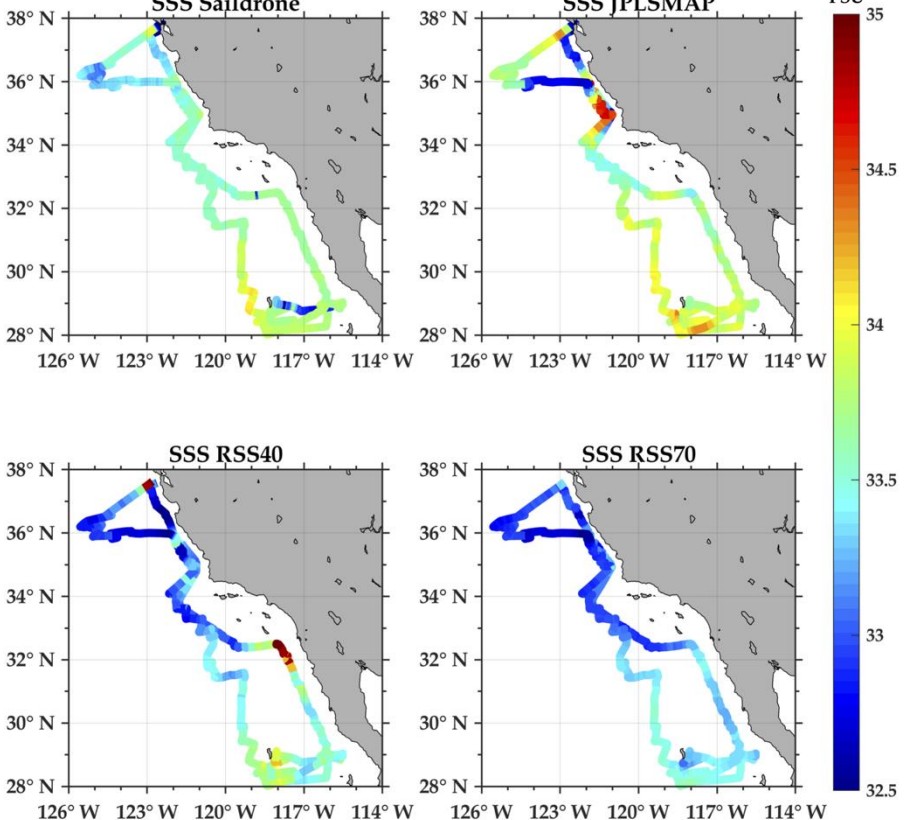

**Figure 2.** Sea surface salinity (SSS) from a Saildrone CTD, JPLSMAP, RSS40, and RSS70.

Figure 3 shows the time-series of SST for Saildrone, MUR, OSTIA, and CMC along the Saildrone track. We deliberately only show results for MUR, OSTIA, and CMC, since results with DMI, K10, and REMSS were found to be similar. Thus, MUR, OSTIA, and CMC were found to be representative of the statistics for the GHRSST Level 4 SST data.

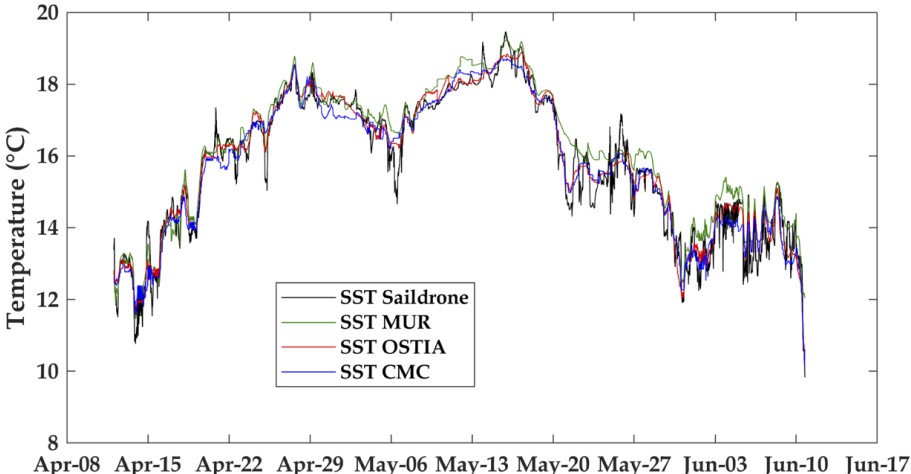

**Figure 3.** SST time-series from Saildrone CTD, MUR, OSTIA, and CMC. MUR, OSTIA, and CMC are co-located to the Saildrone track.

Overall, the GHRSST L4 products captured the dominant features seen in the Saildrone data. There are clear indications though that there are periods of time when the discrepancies were larger. For example, the MUR SST showed the warm bias predominantly during the May time frame, a time period when the Saildrone was further offshore. Understanding these biases is beyond the scope of this work. Possible explanations for such biases could include air–sea interactions and biases in input data, as different sensors are used in the optimal interpolation. In comparison to the SST results (Figure 3), SSS values show larger overall discrepancies with respect to the Saildrone data (Figure 4). Differences of 1 PSU with respect to the Saildrone SSS were seen at different locations along the deployment. A large salty bias in the RSS40 product was seen shortly after mid-April. The JPLSMAP showed a significant salty bias in early June. Biases in the JPLSMAP product, in general, had an opposite sign (i.e., saltier) with respect to the RSS40 and RSS70 biases. Large biases were especially seen during May. The sign of the biases is consistent with the results found in Reference [18] in the Gulf of Mexico. To further identify the possible relationship between the location of the biases and distance from the coast, difference plots were generated similarly to Figures 1 and 2.

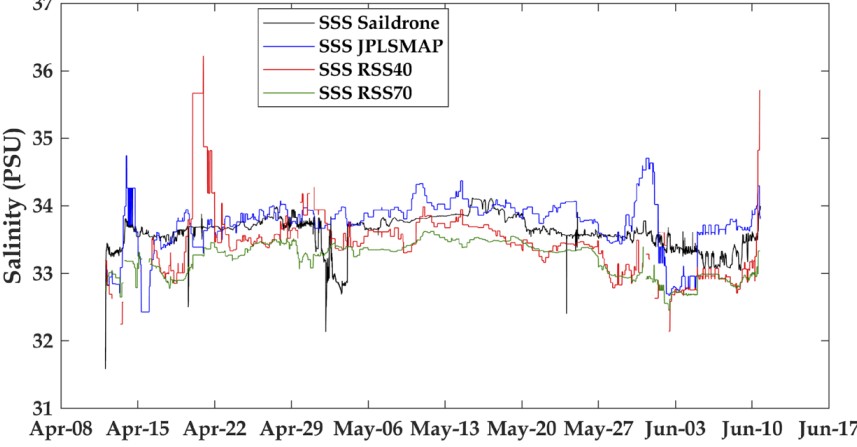

**Figure 4.** SSS time series from Saildrone CTD, JPLSMAP, RSS40, and RSS70.

Figure 5 shows the differences between the different satellite SST products and Saildrone SST along the Saildrone track. The MUR data showed periods of warm biases greater than 1 °C as the Saildrone track was furthest offshore, confirming the spatial representation of Figure 3. Both MUR and OSTIA showed biases close to zero when the deployment was closest to shore. Near-shore differences close to zero indicate that satellite-derived SST is doing well at resolving variability close to the coast.

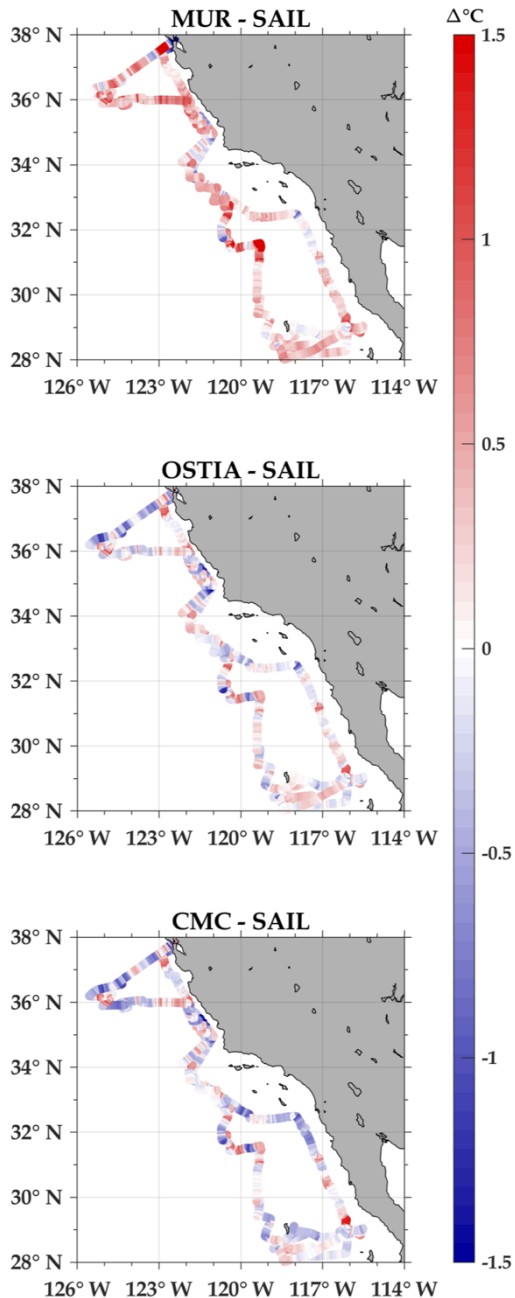

**Figure 5.** Difference plots for each of the remotely sensed SST products and the Saildrone-derived SST.

Figure 6 shows the difference plots for the SSS products. Saltier biases were found for the JPLSMAP product and fresh biases in the RSS40 and RSS70 products. The JPLSMAP product appeared to show reduced biases closer to the coast, while saltier biases were seen further offshore. However, an exception was observed at 36N, where the JPLSMAP had fresh biases greater than 0.5 PSU. The opposite biases from JPL (positive) and RSS (negative) (see Figure 6) seem to suggest that JPL's algorithm overestimated while RSS's algorithm underestimated the land contribution.

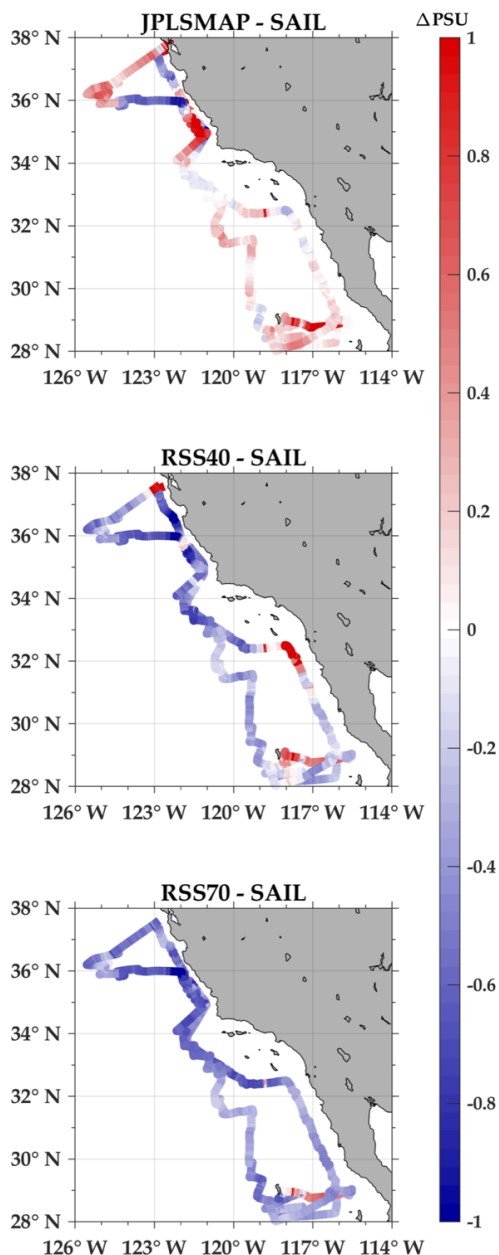

**Figure 6.** Difference plots for each of the remotely sensed SSS products and the Saildrone-derived SSS.

### 3.2. Statistical Comparisons

To summarize the results, statistics were calculated along the entire Saildrone track (Tables 1 and 2). Results are shown for the entire deployment for correlation, bias, RMSD, and the signal-to-noise ratio. Overall, correlations for the six SST products with Saildrone SST range from 0.96 to 0.97, indicating a strong statistical relationship between values from the Saildrone SST and the satellite-derived SST products. These correlations suggest that the remote sensing SST products are resolving a significant amount of the variability associated with the Baja deployment. Biases for the OSTIA, CMC, and DMI products were not significantly different from zero. The OSTIA had the overall minimum RMSD of 0.39 °C, while DMI showed the maximum RMSD of 0.5 °C. The MUR showed an overall warm bias of 0.3 °C (Figure 5). The OSTIA showed the maximum signal-to-noise ratio of 5.0, while the minimum signal-to-noise ratio was found for DMI at 3.4. The high signal-to-noise ratios are consistent with the correlation values mentioned above (0.97). The REMSS, K10, and DMI all showed positive biases with DMI close to zero.

**Table 1.** Bias, RMSD, correlation, and signal-to-noise ratio for MUR, OSTIA, and CMC, with respect to the Saildrone-derived SST. REMSS, K10, and DMI products have been added for comparison.

| Parameter | Bias (°C) | RMSD (°C) | Correlation | Signal-to-Noise Ratio |
|-----------|-----------|-----------|-------------|-----------------------|
| CMC | −0.03 | 0.43 | 0.97 | 4.3 |
| OSTIA | 0.04 | 0.39 | 0.98 | 5.0 |
| MUR | 0.32 | 0.42 | 0.97 | 4.4 |
| REMSS | 0.11 | 0.43 | 0.97 | 4.3 |
| K10 | 0.16 | 0.49 | 0.96 | 3.7 |
| DMI | 0.04 | 0.5 | 0.96 | 3.4 |

**Table 2.** Bias, RMSD, correlation, and signal-to-noise ratio for JPLSMAP, RSS40, RSS70, with respect to the Saildrone-derived SSS.

| Parameter | Bias (PSU) | RMSD (PSU) | Correlation | Signal-to-Noise Ratio |
|-----------|-----------|-----------|-------------|-----------------------|
| JPLSMAP | 0.13 | 0.37 | 0.39 | 1.0 |
| RSS40 | −0.17 | 0.46 | 0.39 | 1.1 |
| RSS70 | −0.39 | 0.22 | 0.57 | 1.1 |

The MUR RMSD value of 0.42 °C is higher than the values between 0.3 °C and 0.4 °C reported in Reference [31]. Considering the mesoscale-to-submesoscale variability associated with the California/Baja coast, the results compare well to other studies. The significance of these results will be discussed further in the next section. Overall, the high resolution of the SST data is contributing to resolving the high spatial variability along the California and Baja coasts.

Because the three satellite-derived salinity datasets (JPLSMAP, RSS40, and RSS70) had a different spatial coverage, co-locations were based on the feature resolution of the products. Therefore, the spatial co-location window (see Methods and Materials) is based on the resolution of the Level 4 SST and Level 3 SSS data. For the SSS data, we averaged all pixels within a 25 km, 40 km, and 70 km distance from the Saildrone sampling.

Table 2 shows the results using all the co-locations for each product. Correlations with Saildrone values were the highest for RSS70 (0.57), while correlations with JPLSMAP and RSS40 were lower (0.39). The RSS70 had the smallest RMSD of 0.23 PSU, while the JPLSMAP and RSS40 had RMSDs of 0.37 and 0.46 PSU, respectively. The JPLSMAP had a salty bias of 0.13 PSU, while the RSS40 and RSS70 products had fresh biases of 0.17 PSU and 0.39 PSU, respectively. Signal-to-noise ratios were all close to 1, significantly lower than found for the SST products. The overall statistics indicate larger differences in the satellite-derived SSS products, compared to SST, with respect to Saildrone.

## 4. Discussion

Differences between the Saildrone observations and satellite-derived SST and SSS products could be explained by the inherent capability of Saildrones to resolve much higher spatial (sub-kilometer) and temporal (sub-daily) scales. Sub-daily here would be defined as all frequencies higher than one day. The average daily variability of the Saildrone SST was 0.38 °C. Overall, this is approximately the mean RMSD between the satellite-derived products and Saildrone SST (Table 1). However, for SSS, the daily variability was 0.10 PSU, which is significantly lower than the RMSDs of 0.3 PSU between the satellite-derived SSS and Saildrone SSS (Table 2). Hence, for SSS, the differences cannot be simply explained by the daily to sub-daily variability of SSS as measured by Saildrone. This is also reflected in the higher signal-to-noise ratios for the SST products when compared to SSS. For SST, the differences can be explained by the unresolved spatiotemporal variability at the daily and sub-kilometer scales. This could also include diurnal variability [41]. To examine the relationship between the satellite-derived parameters and the spatial scales, wavelength spectra and coherences were calculated for each of the SST and SSS products (Figures 7 and 8). The spectra were calculated using the full time-series of the co-located data, but only for the first 500 km. This essentially gave the spectra for the Southward track

of Saildrone, but also at its closest approach to shore. Thus, the plots of the spectra are representative of the approximate sampling of the Saildrone 1 min samples. Based on the feature resolutions of the satellite-derived products, the actual Nyquist periods were 10 km for OSTIA, 2 km for MUR, 20 km for CMC, 120 km for JPLSMAP, 80 km for RSS40, and 140 km for RSS70. Fifty kilometers was chosen for SSS based on the 25 km grid resolution of the SSS data.

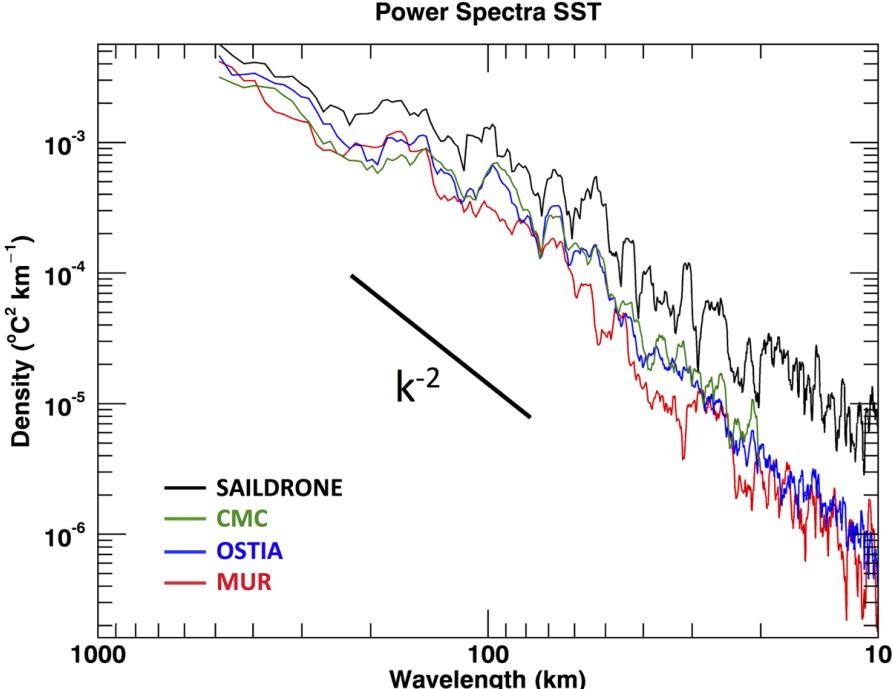

**Figure 7.** Power spectra for the three satellite-derived SST products and Saildrone SST. For the purpose of reference, the $k^{-2}$ slope is overlaid.

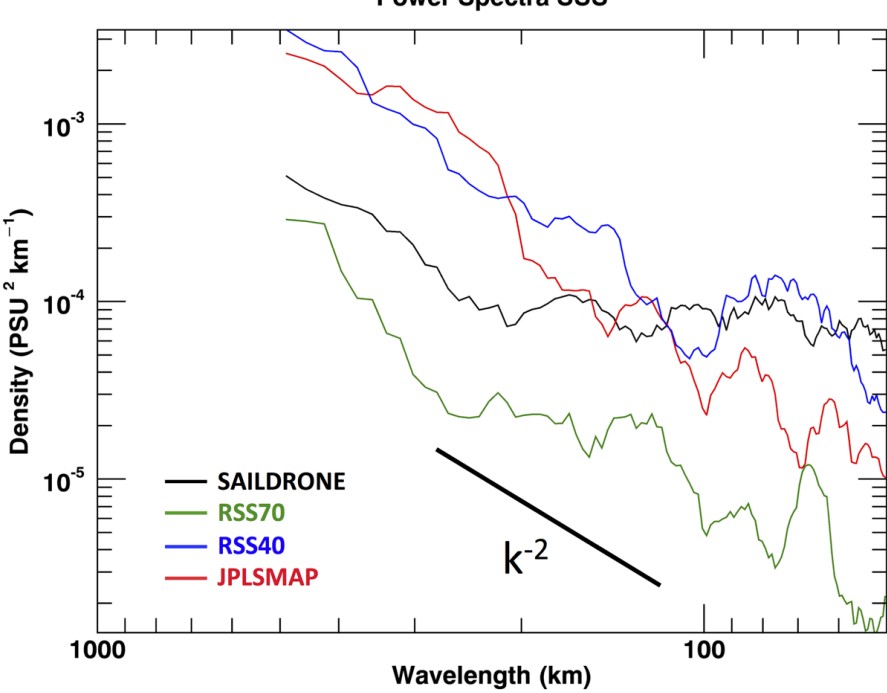

**Figure 8.** Power spectra for the three satellite-derived SSS products and Saildrone SSS. For the purpose of reference, the $k^{-2}$ slope is overlaid.

Comparisons of the MUR, OSTIA, and CMC wavelength spectra with Saildrone indicate that satellite-derived products are resolving a significant part of the spatial variability in the region. Of course, this is for spatial scales greater than 20 km, the approximate Nyquist wavelength associated with the CMC product. Overall, the Saildrone SST showed larger spatial variability, especially at scales smaller than 100 km associated with the mesoscale to submesoscale. The results of the SST spectra were consistent with the large signal-to-noise ratios in the direct comparison between values of the remote sensing SST products and Saildrone SST measurements.

Spectral slopes were calculated for the SST and SSS spectra (see Table 3). Overlaid on Figures 7 and 8 (as a reference) is the spectral slope of $k^{-2}$, which is used as a reference as it is commonly associated with power spectra resolving the mesoscale variability [15]. Table 3 summarizes the spectral slopes, based on the above spectra of the different SST and SSS products. All spectral slopes approximately followed the $k^{-2}$ slope.

**Table 3.** Wavelength spectral slopes.

| Parameter | $(k^{-1})$ |
|:---:|:---:|
| SST Sail | −2.22 |
| SST MUR | −2.11 |
| SST OSTIA | −2.12 |
| SST CMC | −2.12 |
| SST REMSS | −1.92 |
| SST DMI | −2.02 |
| SST K10 | −2.03 |
| SSS Sail | −1.81 |
| SSS JPLSMAP | −1.69 |
| SSS RSS40 | −1.96 |
| SSS RSS70 | −1.91 |

Slopes were derived from 500 km to 20 km for the SST spectra and 500 km to 100 km for the SSS spectra, thus including the mesoscale and a portion of the submesoscale variability. The decision to examine the slope starting at 500 km was based on the approximate north–south distance of the Saildrone deployment. As a reference, Saildrone SST had a spectral slope of $k^{-2.2}$. The OSTIA, CMC, and MUR all had spectral slopes of $k^{-2.1}$. The REMSS, DMI, and K10 also had spectral slopes of approximately $k^{-2}$. The spectral slopes of $k^{-2}$ to $k^{-3}$ were consistent with the kinetic energy spectra for mesoscale to submesoscale variability [15,19]. Reference [15] compared spectral slopes from MODIS Terra and Aqua, as well as an unmanned surface vehicle (Ball Experimental Infrared Radiometer). They found spectral slopes of $k^{-2}$ for wavelengths between 10 m and 100 km, thus encompassing primarily submesoscale variability. Spectral slopes for the REMSS, DMI, and K10 SST were more "white" than MUR, OSTIA, and CMC, but not statistically different. The relative consistency between the SST slopes indicates there were no major differences in the noise level inherent to the products. The SSS spectral slopes were also consistent with mesoscale variability. Spectra at wavelengths of less than 100 km for SSS must be interpreted with caution due to the different Nyquist frequencies of the SSS satellite-derived products. Saildrone, JPLSMAP, RSS40, and RSS70 SSS spectra had a slope of $k^{-1.8}$, $k^{-1.7}$, $k^{-2}$, and $k^{-1.9}$. Both the SST and SSS spectral slopes were consistent with the results reported by Reference [42], using a singularity method. Using data from the SMOS mission and OSTIA, they determined spectral slopes of $k^{-2.4}$ for wavelengths from 10,000 km to 100 km. The overall magnitude of the wavelength spectra is consistent with previous results [15,19,41]. Based on the consistency of the spectral slopes, both the SST and SSS are resolving scales associated with the mesoscale variability. However, the SSS is limited to scales greater than 100 km. This is further confirmed by the coherences.

The next analysis focused on the coherence to determine at what wavelengths the relationships between the satellite-derived products and Saildrone were significant. Figures 9 and 10 show the coherence between the SST and SSS satellite-derived products and Saildrone. The SST shows coherence

between 0.7 and 0.8 for spatial scales greater than 300 km, then a minimum before a local maximum at approximately 100 km (Figure 9). The MUR had the highest coherence value at 300 km, while OSTIA had the highest coherence value at 100 km. Error bars indicate that peaks at 300 km and 100 km are statistically significant. At spatial scales smaller than 30 km, coherences for all the products become statistically insignificant. Thus, at spatial scales associated with the submesoscale variability, the Saildrone SST is likely resolving oceanic structures and processes that are not captured in the satellite-derived products.

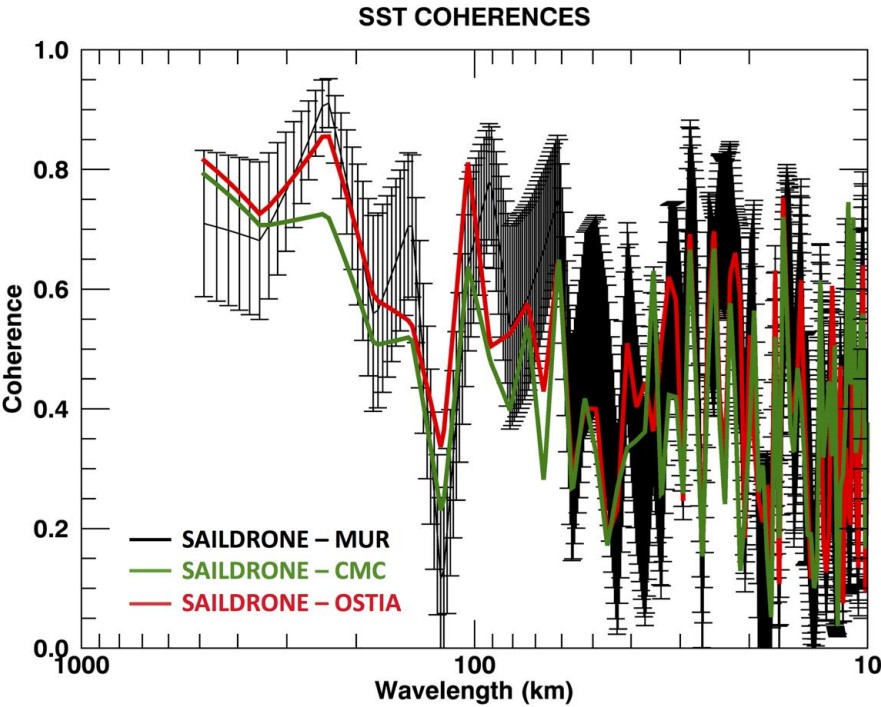

**Figure 9.** Coherences between the Saildrone SST and the SST from the MUR, OSTIA, and CMC products.

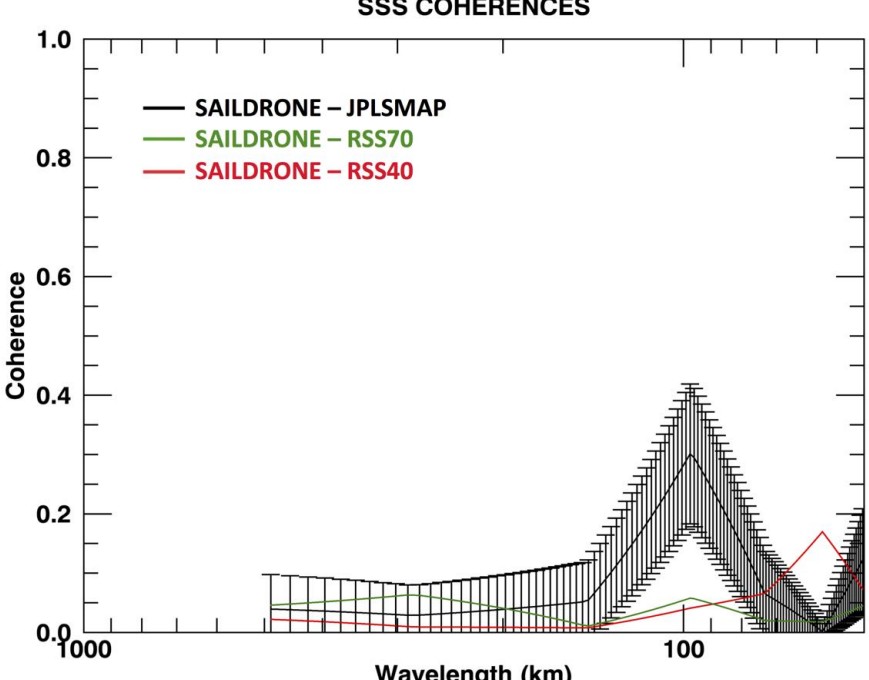

**Figure 10.** Coherences between the Saildrone SSS and SSS from JPLSMAP, RSS40, and RSS70 products.

The SSS coherences showed a statistically significant peak between the JPLSMAP and Saildrone SSS at 100 km (Figure 10). As with SST, all the coherences become statistically insignificant at wavelengths less than 50 km. Again, this indicates that the satellite-derived products are not fully resolving the spatial scales associated with submesoscale variability. Overall, the SST coherences were larger for scales of less than 100 km, but the error bars indicate that appropriate caution be taken in interpreting the significance of these findings. The results indicate that the SSS-derived products were resolving some of the mesoscale variability associated most likely within the California Current System, but not the submesoscale variability. This is, of course, also reflective of the Nyquist wavelength for the satellite-derived SSS products.

The maxima in coherences at 100 km can best be understood by the coastal upwelling in the region [20]. A major difference in the SST and SSS spectra is the lack of coherence in the SSS at spatial scales exceeding 100 km. This is most likely due to the issues of land contamination (see Section 2.1.2). The SSS spectra showed differences among the three products, especially at wavelengths of less than 100 km.

The peaks found at 100 km are very encouraging in showing that both satellite-derived SST and SSS are coherent at scales associated with mesoscale turbulence in the California Current System [18,19]. The upwelling system along the Californian coast, in general, is within 25 km of the coast, but the influence can be much larger. Thus, the maxima in coherence seen at 100 km would be consistent with both the influence of upwelling along the California Coast as well as the inherent resolutions and Nyquist wavelengths of the datasets. The SST spectra showed statistically significant coherences at wavelengths smaller than 100 km, consistent with the higher resolution of the SST data and the ability in resolving smaller spatial scales. However, the error bars indicate that neither the SST nor SSS satellite-derived products were fully resolving the variability associated with the submesoscale.

## 5. Conclusions

This study presents the first known validation of satellite-derived SSS and SST measurements off the California and Baja coasts, using a USV. Overall correlations of Saildrone SST with SST values from MUR, OSTIA, CMC, REMSS, K10, and DMI products exceeded 0.96. The OSTIA and CMC showed biases that were close to zero, with MUR showing warm biases of 0.3°C, RMSD differences of 0.4–0.5 were consistent with other validation studies on regional to global scales. The consistency of RMSD off the California/Baja coasts with global comparisons is promising for applications of high-resolution SST retrievals in coastal regimes. Coherences between MUR, OSTIA, CMC, and the Saildrone SST were close to one at the longer wavelengths with a minimum at approximately 200 km before increasing again at 100 km.

Results for SSS and the comparisons of Saildrone SSS with the JPLSMAP, RSS40, and RSS70 satellite-derived products are encouraging, but not as statistically significant as for SST. Most likely this is due to two issues: (1) the lower spatial resolution of the SSS satellite-derived data; (2) land contamination. Land contamination results when part of the satellite footprint is over land. For both SMAP and SMOS, this occurs at distances less than 100 km from land. Overall, the highest correlation (approximately 0.6) was found between Saildrone SSS and the RSS70 product, while the JPLSMAP and RSS40 products had correlations of 0.4. Results are consistent with the RSS40 product having the least spatial smoothing, thus more noise, but the highest spatial resolution. This is consistent with the lower S/N ratio. The RSS70 product showed the lowest RMSD around 0.2 PSU, while the RSS40 product showed the highest RMSD at approximately 0.46 PSU. Additionally, when RMSD values of satellite minus Saildrone were compared with the daily variability of SSS and SST observations from Saildrone, the RMSD for SST can be explained by the unresolved daily variability. However, for SSS, RMSD values were significantly larger, indicating the increased noise and/or possible land contamination of the satellite-derived SSS products. The results are encouraging, though, that the RMSD of 0.3 PSU was only slightly higher than the RMSD values of global comparisons of 0.2 PSU [3]. Coherences showed a peak at 100 km (same as SST), but became statistically insignificant at wavelengths < 100 km.

With this work, we intended to illustrate the potential of using USVs for validating remotely sensed ocean data in coastal regions. Future work will focus on applications of remote sensing data in challenging regimes while incorporating Saildrone to validate the satellite products and the relationship to mesoscale–submesoscale features. Additionally, the comparisons show that more work needs to be done in the validation of both SST and SSS in coastal regions, with satellite-derived SSS products showing significant differences with respect to Saildrone.

**Author Contributions:** All the authors played a critical part in the preparation of the manuscript. Each author brought their expertise, including scientific expertise in the California Coast and Baja Coast.

**Funding:** This research was supported by NASA Salinity Continuity Program as well as the National Ocean Partnership Program (NOPP) Multi-Sensor Improved Sea Surface Temperature (MISST) Program. J.G.V. and L.E.M. were supported by CICESE and CONACYT, grant #257125, México.

**Acknowledgments:** The research described in this paper was carried out at the Jet Propulsion Laboratory, California Institute of Technology, under a contract with the National Aeronautics and Space Administration (NASA). This research was supported by the NASA Science Utilization of the Soil Moisture Active-Passive Mission Salinity Continuity program as well as the National Ocean Partnership Program. Data for this paper are available at the Physical Oceanography Distributed Active Center (PO.DAAC (http://podaac.jpl.nasa.gov)).

**Conflicts of Interest:** The authors declare no conflict of interest.

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
