# Peer review of "Using Saildrones to Validate Satellite-Derived Sea Surface Salinity and Sea Surface Temperature along the California/Baja Coast"

_remotesensing, doi:10.3390/rs11171964_

Round 1

Reviewer 1 Report

Revision of the draft by Vazquez-Cuervo et al, “Using Saildrones to Validate Satellite-Derived Sea Surface Salinity and Sea Surface Temperature along the California/Baja Coast”
The authors compare the SST and SSS measurements of a saildrone to the measurements derived from data by different satellites.  
The saildrone is a very interesting platform and presumably is easy to be calibrated in terms of conductivity and temperature measurements, therefore publications about their use is very welcome and a step forward to scientific research.
The satellite derived SST data are well described, they are based on data sets from different satellites combining with different near surface temperature in situ data.
The satellite derived SSS data are based solely on the SMAP satellite data.
The intention of the draft is to validate the capability of the satellite derived products to measure the mesoscale or sub-mesoscale variations in SST and SSS in a coastal environment.
Maps of biases and Root mean square differences between the saildrone- and satellite data are shown for estimating the measured values, and spectral density estimates are used for validating the spatial variability captured by the different data sets.
In general, I find the the interpretation of the results unsufficient, they need a better elaboration.
1.    The saildrone data shows several peaks in the evolution of salinity (fig. 4) these peaks are not related to a peak in temperature. The question arises, if the Sail drone data were processed in an adequate way. The peaks should be discussed.
2.    The SSS satellite data are known to suffer from land contamination (LC), even the SMAP data do. There are different approaches to overcome this problem. A discussion of this point is necessary. The processing and treatment of the data is well documented in the references given, however, the relevant aspects of the data processing should be mentioned. E. g., the RSS40 and RSS70 km data set only differ in their spatial resolution? How the JPL70 data is bias corrected? In this context, the difference data could be plotted against the distance from the coast. Otherwise, the different biases from JPL and RSS data have to be explained (negative for RSS and positive for JPL, Fig. 6). In the last chapter, the LC was mentioned, however, not sufficiently discussed. By the way, SMOS suffers stronger from LC than SMAP and as far as 1000 km from land, due to the interferometric method, a fact which also was not mentioned.
3.    Whereas the comparison between the SST data is quite convincing this is not the case for the SSS data (RMSD 0.1 or less in most SST data, against 0.3 or more in the SSS data). Reasons for the discrepancy is not found in using a different colocation method, this second approach (Table 3) can be omitted, reasons have to be found in the measurement technique and the processing of the data. The colocation method is questionable in another aspect: The satellite SSS maps represent a spatial average, and, depending on the product used, also a temporal average of at least one week, whereas the saildrone is a point measurement. This is different from the SST offering a higher resolution and should at least be discussed. Furthermore, the small amount of data from the saildrone is not helpful for a thorough statistics, this, too, should at least be mentioned.
4.    The SSS satellite data have different spatial resolutions of 40, 60 and 70 km. How the spectral density estimates below these spatial scales can be interpreted? And why the RSS70  data show spectral density much lower than the RSS40 data in wavelength > 100 km? JPL (60 km) and RSS40 (40km)  have similar spectral densities at the big scales.
5.    Houreau et al [30] have found spectral slopes of k -2.4 for scales > 100 km, but here the authors try to interprete scales of < 100 km.
6.    Moreover, it is a question if the the saildrone data may be comparable to the satellite data on such a big distance of 500 km, first because it takes the saildrone about one month to cover this distance and second, the trajectory of the sail drone is not a section but is full of turns. Last but not least the number of sections is really small.

This draft poses more questions than it is giving satisfactory results, and in my opinion a validation study should be published in a more elaborated manner. Therefor I don’t recommend publication without a substantial review.

Author Response

Report 1:

May 29, 2019

English language and style

( ) Extensive editing of English language and style required 
( ) Moderate English changes required 
(x) English language and style are fine/minor spell check required 
( ) I don't feel qualified to judge about the English language and style 

Yes

Can be improved

Must be improved

Not applicable

Does the introduction provide   sufficient background and include all relevant references?

( )

( )

(x)

( )

Is the research design   appropriate?

(x)

( )

( )

( )

Are the methods adequately   described?

( )

(x)

( )

( )

Are the results clearly presented?

( )

(x)

( )

( )

Are the conclusions supported by   the results?

(x)

( )

( )

( )

Comments and Suggestions for Authors

This paper evaluates the quality of satellite SST and more recently available SSS products in a coastal context. It is done by comparison with USV measurements, a relatively new technology that I had not seen previously used for satellite validation, which is the main originality of the paper. It provides useful and elaborate statistics on the satellite/in situ comparison, and relate the differences found to the limitation of satellite products (especially SSS) for capturing (sub)mesoscale and near-coast variability. I think this paper deserves publication after minor revisions that I suggest in my comments below.

L67-71 The order the satellite missions are presented implicitly suggests that Aquarius came first, while SMOS was the first mission (still operational), followed by Aquarius and SMAP. Please order chronologically.

Answer. Done.

L73, L84 “Traditional” validation of satellite SSS does not limit to comparison with Argo, the scientific community is well aware of the need to go beyond point measurements. Ship TSG data are also used in cited references [16] and [18] to validate mesoscale and coastal variability in satellite SSS at global scale, or in regional contexts (e.g. Grodsky et al., 2012, GRL; Akhil et al, 2016, IJRS). Also TSG data were used as reference to characterize global SSS submesoscale variability and its implication for satellite SSS comparison with point measurements in a recent paper by Drushka et al. (http://doi.org/10.1175/JPO-D-19-0018.1). I agree that saildrones are a promising technology but it should be mentioned that TSG measurements have been until now the “traditional” way to account for mesoscale variability in satellite validation.

Answer. Done. TSGs are now mentioned.

Part 2.1.1 It may be worth mentioning at which depth salinity is measured with the saildrone. It is presumably shallower than ship TSG measurements which would ease comparison with satellite SSS.

Answer. At 0.6 m depth.  Done.

L120 Can you precise how this temporal resolution translates into spatial resolution, given the  saildrone typical speed?

Answer. Saildrone travels at an average speed of  2 m/s, so the Saildrone covers a distance of 120 m in 1 minute.  Done.

L128 Please make it clear in this first sentence of the paragraph that all 3 products are SMAP-derived. I had to guess that from the web links given later and found confirmation in part 2.2, after initially thinking you would compare SMAP and SMOS products.

Answer. Done.

L172 10 km gridded data

Answer. Done.

L 221-232 This summary of previous sections 2.1.2 and 2.1.3 is useless here and should be suppressed. Section 2.1.2 should start with the sentence in L221 to make it clear all 3 products are SMAP-derived.

Answer. Those sentences were removed. Section 2.1.2 clearly states now that the three salinity data sets are SMAP derived.

L236, 261 Can you explain why the bias, spectra and coherences are apparently not computed for 3 of the 6 SST products?

Answer. The primary reason is that the spectra/coherences for all the SST data sets were very similar.  We decided to plot only the three SST data sets to keep the graphs simple and easily readable. The correlations and biases are listed in the table.  The spectral slopes for all six products are also listed in the table.

Equation (3) Is the SDROMEAN the saildrone SST/SSS averaged over the whole Baja deployment or a running average at a given spatial scale?

Answer. The SDROMEAN is the mean over the entire deployment.

L294 This interpretation of the figure is out of place in the legend and should be moved to the main text.

Answer. Done

L297 Do you mean they show differences with Similar one each other, or all 3 similar to another SST product shown on Figure 3?

Answer. They are all similar with respect to the comparisons with Saildrone.

L325 Note that they show a saltier bias along a meridional section at 29°N, where the JPLSMAP bias is maximum too.

Answer. Based also on comments from other reviews we redid the co-locations so each Saildrone point was co-located with the satellite derived product. Additionally, window sizes for the salinity product were based on 40km (RSS40), JPLSMAP (25km) and 70km (RSS70).  The primary conclusion about the overall biases is consistent, but more structure is seen, especially along the coast.

Figure 6 How can you compute a bias for sections where no satellite SSS is available, according to Figure 2 (coastal section between 29-32°N for RSS70, around 35°N for RSS40 and RSS70)?

Answer: We apologize for the confusion on this point. It was error on our part. We now applied consistently the co-location strategy based on box average at each Saildrone sample. This is now described in the text. This now allows us to fill in all the Saildrone samples with matchups/co-location.  

L332 Why do you only comment on the statistics for 3 SST products while all 6 are now included in the table?

Answer.  We have now added some statements commenting on all the SST products.

L347 RMSD value is 0.46°C for MUR SST according to table 1, therefore outside of the 0.3-0.4°C range and not consistent with the cited reference. Please revise accordingly.

Answer. Statement was reworded.  Now explicitly says, “higher”.

L348 comparisons done by [24]

Answer. Done.

L357 Can you clarify which impact on the statistics (degrade/improve?) you expect for JPLSMAP and why?

Answer. This section has now been rewritten as we used only one statistic. We used a co-location strategy that implemented different window sizes based on the inherent smoothing in the three products. This filled in the entire saildrone deployment with all the satellite derived SSS products. Table 3 was thus removed.

L360-361 The correlation numbers are taken form the RMSD column and the RMDS numbers from Table 3 instead of Table 2, apparently. Please correct.

Answer. Apologies we redid this section to make things more clear.

L369 You choose the highest correlation for JPLSMAP but the lowest for RSS40 among Tables 2 and 3. This seems not fair. I think that “overall” correlation should be taken from Table 2.

Answer. Yes, this has now been changed. Based on the new co-location strategy the highest correlation is seen the RSS70km. This would be consistent with the product being the smoothest.

L371,373 I do not really see the need for repeating the bias and RMDS numbers already given a few lines above, and rounding them (sometimes incorrectly). Please correct. Actually this paragraph summarizing of statistics is repeated in the second paragraph in the conclusions so it could be suppressed here.

Answer. Done.

L431 Coherence is rather around 0.7-0.8 for spatial scales greater than 300 km in Figure 9.

Answer. Done

L466 This study presents...

Answer. Done.

L468 and DMI products exceed 0.96

Answer. Done

L479-484 Please check your statistics with Tables 2 (see comments above for L369, 371, 373).

Answer. This section has been rewritten as tables 2-3 have now been consolidated with the consistent co-location strategy. The co-location strategy allows for the RSS40km and RSS70km products to have matchups over the near complete Saildrone track.  

Submission Date

15 May 2019

Date of this review

28 May 2019 18:40:55

Reviewer 2 Report

The paper by Vazquez-Cuervo et al. examines the accuracy of the satellite-derived SST and SSS products using saildrone measurements along the California/Baja coast with a particular goal to evaluate the impact of mesoscale to submesoscale variability. The topic is particularly interesting as the coastal areas are difficult to evaluate due to lack of adequate in-situ data and thus such evaluations are very rare. From this perspective, the results might be interesting and worthy of publication. However, I have some serious concerns regarding both the clarity of the writing and interpretation of the results, for which I request a substantial revision.    

Major:

Eq. 3: What does ‘SDROMEAM’ mean? If it is the mean over the whole record (~850 km long in the North-South direction) than it has to be stated that the estimated signal-to-noise ratio is for the largest scale resolved by saildrone measurements. Technically, the signal-to-noise ratio is a function of length scale which can be estimated from the spectra presented in the Discussion section.

Fig. 2 shows gaps in satellite SSS near the coast, which magically disappear in Fig 6 (the differences between satellite SSS and saildrone). Please clarify how the data gaps were filled.

One can easily estimate the sub-footprint or sub-grid variability from the saildrone data, e.g STD in a moving window, let’s say 40-km wide. The saildron track is ~2000 km long which gives approximately 2000/40=50 quasi-independent estimates. Examples how to do this type of analysis can be found in, e.g.., Lagerloef and Delcroix (1999), OceanObs99 paper. Such estimates would strengthen the paper immensely as “evaluating the impact of mesoscale to submesoscale variability” on validating satellite SST and SSS data is one of the goals of the paper.

In the Discussion section I got lost, quite frankly. A little bit of detail how the spectra were computed would be extremely useful here, given the fact that the saildrone track is not a straight line. For SST products, the Nyquist wavelength is ~20 km (line 399). Everything below this scale in Fig. 7 is interpolation noise. For SSS products, the Nyquist wavelength is ~50 km (2xgrid size), so only a small portion of the spectrum in Fig. 8 (actually to the left of where the k^-2 curve is drawn) reflects the signal. Further, line 459: “…the Nyquist wavelength would be 120 km for the JPLSMAP product, 80 km for RSS40, and 140 km for RSS70”, which makes even larger portions of the spectra in Fig 8 invalid. In these circumstances, I believe, fitting the slopes and concluding that they are all “…consistent with mesoscale to submesoscale variability” (line 420) and that “both the SST and SSS are resolving scales associated with mesoscale and submesoscale variability (line 426) is a bit premature.  At longer scales (~100-500 km) JPLSMAP and RSS40 SSS spectra have more power than the saildron, which points to large correlated errors or biases in the satellite data. This may also help explain the lack of coherence in Fig 10.

Minor:

Lines 141: “The 70 km product should be smoother.” “is” smoother?

Lines 214-215: SMAP and GHRSST already abbreviated in the Introduction sec.

Lines217-219: “The primary goal here was not to validate each of the satellite derived SST and SSS products individually… thus validating the satellite-derived products….”  This sentence is difficult to understand. Please clarify.

Lines 219-232: Repetition. The data have already been described in sec 2.1.1-2.1.3.

Lines 238-250: “…a nearest-neighbor approach” for daily satellite data means interpolating satellite maps into saildrone locations and times with the method “nearest”.  Describing the collocation window as some L plus/minus 1 day creates an impression that each saildrone data point may technically have 3 satellite collocations.

Fig. 4: Please make the y-axis, let’s say, from 31.5 to 36.6, instead of 28-38 (psu), to emphasize the differences. The spike at the end of the record is not important to show.

Line 451: “This is most likely due to the data gaps…” The JPLSMAP SSS record does not have any gaps (Fig 2), yet it shows low coherence.

Author Response

Report 2:

The paper by Vazquez-Cuervo et al. examines the accuracy of the satellite-derived SST and SSS products using saildrone measurements along the California/Baja coast with a particular goal to evaluate the impact of mesoscale to submesoscale variability. The topic is particularly interesting as the coastal areas are difficult to evaluate due to lack of adequate in-situ data and thus such evaluations are very rare. From this perspective, the results might be interesting and worthy of publication. However, I have some serious concerns regarding both the clarity of the writing and interpretation of the results, for which I request a substantial revision.

Major:

Eq. 3: What does ‘SDROMEAM’ mean? If it is the mean over the whole record (~850 km long in the North-South direction) than it has to be stated that the estimated signal-to- noise ratio is for the largest scale resolved by saildrone measurements. Technically, the signal-to-noise ratio is a function of length scale which can be estimated from the spectra presented in the Discussion section.

Answer. We have now added explicitly that the signal to noise ratio is for the entire ratio. SDROMEAN is for the entire deployment.  The following statement was added: “Thus, as defined by equation 3) SDROMEAN is the mean over the entire deployment and thus the signal to noise ratio is representative over the entire spatial scale. “ We feel, as the goal of the paper was to present an initial comparison, that the methodology for using the spectra to determine signal to noise ratios at all scales would best be reserved for future work.  Additionally, accidently figures,1,2,5,6 used the co-location strategy based on daily averages of Saildrone. This was not consistent with  Figures 3,4. These figures have been updated to reflect the co-location strategy based on using the full sampling of Saildrone. A detailed description of the co-location/matchup strategy is now included in the Methods section. The authors apologize for this error.

.

Fig. 2 shows gaps in satellite SSS near the coast, which magically disappear in Fig 6 (the differences between satellite SSS and saildrone). Please clarify how the data gaps were filled.

Answer. Please see the new Figure 1,2,5,6 which were accidently not updated to reflect the co-location strategy using the full sampling of Saildrone.  This is now explained in the manuscript. The co-location strategy now fills in almost the entire time series. Statistics for table 2 were recalculated based on the co-location strategy using the full sampling of the Saildrone time series. We apologize for the confusion and not updating the figures.

One can easily estimate the sub-footprint or sub-grid variability from the saildrone data, e.g STD in a moving window, let’s say 40-km wide. The saildron track is ~2000 km long which gives approximately 2000/40=50 quasi-independent estimates. Examples how to do this type of analysis can be found in, e.g.., Lagerloef and Delcroix (1999), OceanObs99 paper. Such estimates would strengthen the paper immensely as “evaluating the impact of mesoscale to submesoscale variability” on validating satellite SST and SSS data is one of the goals of the paper.

Answer. We present an estimate of the “subgrid”  variability for SSS and SST by simply getting an average of the “daily” Saildrone SSS and SST. One of the conclusions of the paper is that for SST the subgrid variability can explain the differences between the Saildrone derived SST and the satellite derived products.  This was not the case for SSS. The idea of doing a more detailed analysis of gradients and the subgrid scale variability we hope to do a followup paper focusing on gradients from the products. We wanted to maintain the focus of this original paper on the validation of the products. We thank the reviewer for the comments and the reference paper and hope to implement in a followup paper.  

In the Discussion section I got lost, quite frankly. A little bit of detail how the spectra were computed would be extremely useful here, given the fact that the saildrone track is not a  straight line. For SST products, the Nyquist wavelength is ~20 km (line 399). Everything below this scale in Fig. 7 is interpolation noise. For SSS products, the Nyquist wavelength is ~50 km (2xgrid size), so only a small portion of the spectrum in Fig. 8 (actually to the left of where the k^-2 curve is drawn) reflects the signal. Further, line 459: “...the Nyquist wavelength would be 120 km for the JPLSMAP product, 80 km for RSS40, and 140 km for RSS70”, which makes even larger portions of the spectra in Fig 8 invalid. In these circumstances, I believe, fitting the slopes and concluding that they are all “...consistent with mesoscale to submesoscale variability” (line 420) and that “both the SST and SSS are resolving scales associated with mesoscale and submesoscale variability (line 426) is a bit premature. At longer scales (~100-500 km) JPLSMAP and RSS40 SSS spectra have more power than the saildron, which points to large correlated errors or biases in the satellite data. This may also help explain the lack of coherence in Fig 10.

Answer. Thank you. We have rewritten parts of the section describing the spectra. We have redone the graphs showing the spectra based on the suggestions. For the SST spectra is now shown for 500km to 10km (Nyquist wavelength for OSTIA).  For the SSS spectra we replotted the spectra from 500km to 50km. Fifty kilometers is representative of the Nyquist wavelength for the 25km gridding of the SSS products.  We only used the first 500km of the deployment because the the Saildrone track has both the southward and northward track, thus not representative of the 1000km spatial scale.

Lines 141: “The 70 km product should be smoother.” “is” smoother?

Answer. Done

Lines 214-215: SMAP and GHRSST already abbreviated in the Introduction sec.

Answer. Done.

Lines217-219: “The primary goal here was not to validate each of the satellite derived SST and SSS products individually... thus validating the satellite-derived products....” This sentence is difficult to understand. Please clarify.

Answer. This was rewritten and clarified with the last sentence in the Introduction.

Lines 219-232: Repetition. The data have already been described in sec 2.1.1-2.1.3.

Answer. Done.

Lines 238-250: “...a nearest-neighbor approach” for daily satellite data means interpolating satellite maps into saildrone locations and times with the method “nearest”. Describing the collocation window as some L plus/minus 1 day creates an impression that each saildrone data point may technically have 3 satellite collocations.

Answer. Thank you. This iwas a major revision of the paper. The methods section now has been updated to include detailed  paragraph on the co-location/matchup strategy.

Fig. 4: Please make the y-axis, let’s say, from 31.5 to 36.6, instead of 28-38 (psu), to emphasize the differences. The spike at the end of the record is not important to show.

Answer. Thank you. Figure 4 has been updated.

Line 451: “This is most likely due to the data gaps...” The JPLSMAP SSS record does not have any gaps (Fig 2), yet it shows low coherence.

Answer. This issue of the data gaps has been resolved. Basically figure 2 and figure 6 were regenerated using the co-location strategy at the full resolution of the Saildrone sampling. Figure 4 was also regenerated. The co-location strategy is also now explained more clearly. The following was added to the “Methods and Materials”: section.  “For both SST and SSS satellite products, the co-location with the Saildrone data was conducted by averaging the values of all pixels located within d km from the Saildrone measurement for the same day.d has been selected as the spatial resolution of a given satellite product. Therefore, the spatial co-location window was 1 km for MUR, 5 km for OSTIA and 25 km for JPLSMAP SSS for example. It should be mentioned that co-location in time was restricted to same day measurements regardless of the smoothing used in the Level 3 (SSS) or Level 4 (SST) product. In fact, SMAP SSS products were produced daily but consisted of 8-day running means. These co-location criteria are applied throughout the study, insure that more that one pixel is used for the comparison thus reducing any potential impact of noise in the satellite dataset. Co-locations were done for each Saildrone sample to not degrade the resolution of the Saildrone sampling.”

Reviewer 3 Report

This paper evaluates the quality of satellite SST and more recently available SSS products in a coastal context. It is done by comparison with USV measurements, a relatively new technology that I had not seen previously used for satellite validation, which is the main originality of the paper. It provides useful and elaborate statistics on the satellite/in situ comparison, and relate the differences found to the limitation of satellite products (especially SSS) for capturing (sub)mesoscale and near-coast variability. I think this paper deserves publication after minor revisions that I suggest in my comments below.

L67-71 The order the satellite missions are presented implicitly suggests that Aquarius came first, while SMOS was the first mission (still operational), followed by Aquarius and SMAP. Please order chronologically.

L73, L84 “Traditional” validation of satellite SSS does not limit to comparison with Argo, the scientific community is well aware of the need to go beyond point measurements. Ship TSG data are also used in cited references [16] and [18] to validate mesoscale and coastal variability in satellite SSS at global scale, or in regional contexts (e.g. Grodsky et al., 2012, GRL; Akhil et al, 2016, IJRS). Also TSG data were used as reference to characterize global SSS submesoscale variability and its implication for satellite SSS comparison with point measurements in a recent paper by Drushka et al. (http://doi.org/10.1175/JPO-D-19-0018.1). I agree that saildrones are a promising technology but it should be mentioned that TSG measurements have been until now the “traditional” way to account for mesoscale variability in satellite validation.

Part 2.1.1 It may be worth mentioning at which depth salinity is measured with the saildrone. It is presumably shallower than ship TSG measurements which would ease comparison with satellite SSS.

L120 Can you precise how this temporal resolution translates into spatial resolution, given the  saildrone typical speed?

L128 Please make it clear in this first sentence of the paragraph that all 3 products are SMAP-derived. I had to guess that from the web links given later and found confirmation in part 2.2, after initially thinking you would compare SMAP and SMOS products.

L172 10 km gridded data

L 221-232 This summary of previous sections 2.1.2 and 2.1.3 is useless here and should be suppressed. Section 2.1.2 should start with the sentence in L221 to make it clear all 3 products are SMAP-derived.

L236, 261 Can you explain why the bias, spectra and coherences are apparently not computed for 3 of the 6 SST products?

Equation (3) Is the SDROMEAN the saildrone SST/SSS averaged over the whole Baja deployment or a running average at a given spatial scale?

L294 This interpretation of the figure is out of place in the legend and should be moved to the main text.

L297 Do you mean they show differences with  Similar one each other, or all 3 similar to another SST product shown on Figure 3?

L325 Note that they show a saltier bias along a meridional section at 29°N, where the JPLSMAP bias is maximum too.

Figure 6 How can you compute a bias for sections where no satellite SSS is available, according to Figure 2 (coastal section between 29-32°N for RSS70, around 35°N for RSS40 and RSS70)?

L332 Why do you only comment on the statistics for 3 SST products while all 6 are now included in the table?

L347 RMSD value is 0.46°C for MUR SST according to table 1, therefore outside of the 0.3-0.4°C range and not consistent with the cited reference. Please revise accordingly.

L348 comparisons done by [24]

L357 Can you clarify which impact on the statistics (degrade/improve?) you expect for JPLSMAP and why?

L360-361 The correlation numbers are taken form the RMSD column and the RMDS numbers from Table 3 instead of Table 2, apparently. Please correct.

L369 You choose the highest correlation for JPLSMAP but the lowest for RSS40 among Tables 2 and 3. This seems not fair. I think that “overall” correlation should be taken from Table 2.

L371,373 I do not really see the need for repeating the bias and RMDS numbers already given a few lines above, and rounding them (sometimes incorrectly). Please correct. Actually this paragraph summarising of statistics is repeated in the second paragraph in the conclusions so it could be suppressed here.

L431 Coherence is rather around 0.7-0.8 for spatial scales greater than 300 km in Figure 9.

L466 This study presents...

L468 and DMI products exceed 0.96

L479-484 Please check your statistics with Tables 2 (see comments above for L369, 371, 373).

p { margin-bottom: 0.1in; line-height: 120%; }a:link { }

Author Response

May 29, 2019

Report 3

Open Review

English language and style

( ) Extensive editing of English language and style required 
( ) Moderate English changes required 
(x) English language and style are fine/minor spell check required 
( ) I don't feel qualified to judge about the English language and style 

Yes

Can be   improved

Must be   improved

Not   applicable

Does the   introduction provide sufficient background and include all relevant   references?

( )

(x)

( )

( )

Is the   research design appropriate?

( )

( )

(x)

( )

Are the   methods adequately described?

( )

( )

(x)

( )

Are the   results clearly presented?

( )

( )

(x)

( )

Are the   conclusions supported by the results?

( )

( )

(x)

( )

Comments and Suggestions for Authors

Revision of the draft by Vazquez-Cuervo et al, “Using Saildrones to Validate Satellite-Derived Sea Surface Salinity and Sea Surface Temperature along the California/Baja Coast”

The authors compare the SST and SSS measurements of a saildrone to the measurements derived from data by different satellites.  The saildrone is a very interesting platform and presumably is easy to be calibrated in terms of conductivity and temperature measurements, therefore publications about their use is very welcome and a step forward to scientific research. The satellite derived SST data are well described, they are based on data sets from different satellites combining with different near surface temperature in situ data. The satellite derived SSS data are based solely on the SMAP satellite data. The intention of the draft is to validate the capability of the satellite derived products to measure the mesoscale or sub-mesoscale variations in SST and SSS in a coastal environment. Maps of biases and Root mean square differences between the saildrone- and satellite data are shown for estimating the measured values, and spectral density estimates are used for validating the spatial variability captured by the different data sets. 

In general, I find the the interpretation of the results unsufficient, they need a better elaboration.

1.    The saildrone data shows several peaks in the evolution of salinity (fig. 4) these peaks are not related to a peak in temperature. The question arises, if the Sail drone data were processed in an adequate way. The peaks should be discussed.

Answer. The largest peaks (fresh water) are associated with periods of time when Saildrone is leaving or returning to port. Other peaks (fresher) are associated at times when Saildrone is near land. The issue of the Saildrone accuracy was addressed with the following paragraph that was added: “The accuracy of Saildrone USV salinity observations was examined using nearby ship observations 1-10 May 2015 [Cokelet et al., 2015]. The root-mean-square (RMS) salinity difference was found to be 0.01 PSS78. A longer two-month validation was completed in 2018, and all sensors, except the skin SST and barometric pressure where found to be operating within manufacturer specifications [Gentemann et al. (2019), submitted].” Figure 4 was regenerated with a filter applied to remove the spikes.

2.    The SSS satellite data are known to suffer from land contamination (LC), even the SMAP data do. There are different approaches to overcome this problem. A discussion of this point is necessary. The processing and treatment of the data is well documented in the references given, however, the relevant aspects of the data processing should be mentioned. E. g., the RSS40 and RSS70 km data set only differ in their spatial resolution? How the JPL70 data is bias corrected? In this context, the difference data could be plotted against the distance from the coast. Otherwise, the different biases from JPL and RSS data have to be explained (negative for RSS and positive for JPL, Fig. 6). In the last chapter, the LC was mentioned, however, not sufficiently discussed. By the way, SMOS suffers stronger from LC than SMAP and as far as 1000 km from land, due to the interferometric method, a fact which also was not mentioned. 

Answer. We have added the following to the paper for more detail on the land corrections: “[lN1] Land contamination (LC) is the leakage of energy from land surface into the radiometer receiver through side lobes or partially through the main lobe of SMAP radiometer. In fact, SMOS suffers stronger from LC than SMAP and as far as 1000 km from land, due to the interferometric measurement method. Warmer land temperature mixed with ocean signature causes large error in salinity retrieval near land. The so-called land correction is a step in the retrieval algorithm to remove the land contribution from radiometer measured brightness temperature (TB) before to be used for ocean salinity retrieval. The teams at JPL and RSS developed the land correction scheme using different approaches. Basically, JPL’s land correction is derived from SMAP data itself [Tang et al., 2018], while RSS’ land correction is based on simulated land brightness temperature [Meissner et al., 2018]. The opposite biases from JPL (positive) and RSS (negative) (Fig.6) seem suggest JPL algorithm overestimates while RSS algorithm underestimates the land contribution.

3.    Whereas the comparison between the SST data is quite convincing this is not the case for the SSS data (RMSD 0.1 or less in most SST data, against 0.3 or more in the SSS data). Reasons for the discrepancy is not found in using a different colocation method, this second approach (Table 3) can be omitted, reasons have to be found in the measurement technique and the processing of the data. The colocation method is questionable in another aspect: The satellite SSS maps represent a spatial average, and, depending on the product used, also a temporal average of at least one week, whereas the saildrone is a point measurement. This is different from the SST offering a higher resolution and should at least be discussed. Furthermore, the small amount of data from the saildrone is not helpful for a thorough statistics, this, too, should at least be mentioned.

Answer. We completely agree that the differences in SSS raise more issues. But this type of study is useful for presenting the issues so that they can be used by data providers in reprocessing efforts. Although the data sample from Saildrone is only 2 months, it still provides a valuable comparison for understanding the differences in the data sets, particularly the SSS products. The comparison is also useful in showing the SST products are doing a good job in reproducing the variability off the California/Baja Coasts, as indicated by the 0.97 correlation. Table 3 has been omitted.  Figures 5 and 6 were derived to show how the spatial differences vary along the Saildrone track. The figures can be used to identify areas where the biases are larger. For example, biases are clearly seen in all three products closer to land.  Additionally, the differences in the sign of the biases between the three SSS products are clearly seen.  We have added several sentences describing the difference in the land contamination and flagging between the RSS and JPL products. The primary difference is that the JPL product uses a land climatology to derive a look table.  Table 3 has now been eliminated as we made sure to use the consistent co-location strategy. This has now been explained in the text

4.    The SSS satellite data have different spatial resolutions of 40, 60 and 70 km. How the spectral density estimates below these spatial scales can be interpreted? And why the RSS70 data show spectral density much lower than the RSS40 data in wavelength > 100 km? JPL (60 km) and RSS40 (40 km)  have similar spectral densities at the big scales. 

Answer. The reason the spectra were shown to 1 km was that the co-location was based on the actual sampling of the Saildrone USV, approximately 120 meters. We have redone the figures showing the spectra with a cutoff at the more appropriate Nyquist wavelength. For example, for SST, based on the OSTIA grid resolution of 5 km, the spectra were cutoff at 10 km. The CMC SST spectra Is only shown to 20 km, the Nyquist wavelength associated with the CMC 10 km grid spacing.  For SSS the spectra were cutoff at 50 km, based on the grid resolution of the SSS data. The reviewer is absolutely correct that it doesn’t make any sense to interpret the high wavelengths.  But we completely agree that this is confusing. We now redid the figures showing the spectra with proper cutoffs for the Nyquist Wavelengths.

5.    Houreau et al [30] have found spectral slopes of k -2.4 for scales > 100 km, but here the authors try to interprete scales of < 100 km.

Answer. The primary issue of showing the spectra was to indicate that slopes were consistent with what is known about the mesoscale variability.  We have rephrased that sentence. Obviously the SSS data cannot completely resolve scales < 100 km. The coherence plot for SSS, when cutoff properly, clearly shows a spectral peak at 100 km but no significant coherence at scales < 100 km.  We redid the figures showing the spectra to be consistent with the Nyquist Wavelengths.

6.    Moreover, it is a question if the the saildrone data may be comparable to the satellite data on such a big distance of 500 km, first because it takes the saildrone about one month to cover this distance and second, the trajectory of the sail drone is not a section but is full of turns. Last but not least the number of sections is really small. 

Answer. This is a very valid point. The spectra were of course calculated assuming a synoptic scale. However, the high correlation of the SST data indicates that, overall, the satellite data is resolving the scales associated with the dynamics of the region. This includes of course the California Current which dominates the region over several 100 km. The primary issue is that the 2 months is still a short time scale compared to the seasonal to decadal time scales that affect both the upwelling and the California Current System.

This draft poses more questions than it is giving satisfactory results, and in my opinion a validation study should be published in a more elaborated manner. Therefoer I don’t recommend publication without a substantial review.

The authors believe this should be considered a first study in showing the application of the new Saildrone technology, and satellite data, in a coastal region. The study shows that the results are consistent with the known mesoscale variability of the region.  It also shows that the GHRSST L4 SST products are doing an excellent job of reproducing the variability of the region, as is indicated by the high correlation and the spectral slopes. The results show that, yes, using SSS in coastal regions needs to be done with caution. Additionally, it shows that SSS still needs improvements in coastal regions, where land contamination is a primary issue.  We believe these results are important, as Saildrone provides a unique opportunity for validating both SST and SSS in a coastal region. We thank the reviewer for clearly pointing out issues with the manuscript which we hope, in addressing, have improved the manuscript. We hope the clarification of the co-location/matchup methodology, along with the figures, has made the results clearer.

We would also like to clarify that the point of the manuscript was not to do a full validation of the SST and SSS data sets, but to present a new technology (Saildrone) that has potential for improving our understanding of satellite derived SST and SSS in coastal regions and their applications. The paper is intended as a first step in using this promising new technology.  

Submission Date

15 May 2019

Date of this review

28 May 2019 14:53:52

 [lN1]Falta la comilla que cierra. Donde deberia ir?

Reviewer 4 Report

This paper does a comparison of data from a saildrone mission in spring 2018 to the coast of California and Baja California with satellite SSS and SST measurements. The authors compute RMS differences, bias, signal/noise and correlation. They also compute wavenumber spectra of each measurement. This is a good use for the saildrones, and this kind of comparison makes sense. However, there are a lot of issues with the details of the calculations described in the paper that make it problematic. For reasons described below, the spectra are especially a sticky issue, though more for SSS than SST. The other problem is the description of how the comparisons are carried out. Maybe if they were better described, and I understood better how they were done this could be resolved.

Lines 114-126 contain some problematic plagiarism. I am hoping this is inadvertent on the part of the authors and not repeated in any other parts of the paper.

Detailed comments:

Lines 43-44. Technically argo floats are not "buoys". Also, this line repeats on lines 73-74.

Line 59. What stripes?

Line 98. This contradicts line 95, which says SSS is available as an L4 product.

Line 112. "...acoustic Doppler..." Are the ADCP data going to be used in this study? How about the air temperature, etc.? If not, the authors should skip mention of these data to reduce confusion.

Lines 114-126. This text repeats nearly exactly text that appears at the URL given in line 125, so there should be quotation marks with a reference. Even better, delete this entire paragraph, including the URL, and replace with a reference to [21].

Lines 132-133. If the measurement has an "inherent spatial resolution" (what does that mean?) of 60 km, what is the justification for gridding at 25 km? Also, when the authors say "All the datasets...", do they mean the three given in the previous paragraph? If so, move this sentence to the previous paragraph.

Line 136. Following this link gives a page with this message at the top:

"Please Note: This dataset is retired and will not be maintained in the future. However, this dataset is displayed for archive purposes and may not reflect the most updated information."

The authors should use a more current dataset, and one that will not become unavailable in the  near future. Also, the authors are better off pointing to a DOI in a reference, e.g. reference [21]. (Ditto lines 143, 145, 159, etc.)

Lines 138 and following. This is all very garbled. Again, the authors are taking a 40 km and a 70 km product and gridding it at 25 km (I think they mean 0.25 degrees, which is *not* 25 km). By "The rationale for the two products", do the authors mean the rationale for the 40 km one? Lines 139 and 140 repeat each other. Line 141, smoother than what? The authors need to describe briefly what the real difference is between the 40 and 70 km products.

Line 148. This link is dead. I think PO.DAAC stopped supporting ftp. Since I cannot get to the documents this link is supposed to access, I cannot verify if the ATBD and user guides for the RSS datasets are available at the same location. The authors should do this.

Line 172. What is the "forward stream"? Also "10 km".

Line 173. So this study only uses the 10 km data?

Section 2.1.3. It's not clear why it is necessary to use so many different SST products. Maybe it will be clearer later. Also, line 150 indicates that three products are being used, whereas I count 6.

Lines 211-225. As indicated from the comments below, this paragraph is not comprehensible and needs to be re-written from scratch.

Lines 217-219. Huh? It looks like the different products *are* being evaluated individually.

Line 219. What process?

Line 220. "calculations were derived"? What does this mean?

Lines 221-225. The abbreviations have already been given, as well as access URLs. Most of this can be deleted. (Ditto lines 226-232)

Line 239. So the criterion was different for the other SST datasets?

Line 239. "nearest-neighbor approach" As the saildrones are sampled at one minute intervals (line 120), whereas the satellites are averaged onto one day space-time grid points, this nearest neighbor approach would take the closest one minute saildrone sample (closest in space or in time or in some weighted combination?) to the daily gridded satellite value. This seems like quite a mismatch in time and/or space scales. Perhaps the authors averaged the saildrone data in some way that was not specified in section 2.1.1. I am guessing that each one-minute saildrone sample was compared to a gridded satellite value. This would mean potential large changes in the comparison as the saildrone crossed from one day into another. The details of how this comparison was carried out are very important to the results, so this imprecision needs to be cleared up.

Lines 258-262. Again, the details matter here. Wavenumber spectra? If so, such spectra would normally need to be computed from a linear spatial series or a box. In order to interpret the spectra, we need more information about how they are computed. Almost none is given. For instance, the saildrones are mixing spatial and temporal variability, especially at long wavelengths. For SSS, the satellite is gridded at ~25 km intervals, with a total size of the box being ~500 km north-south. This means 20 samples from which to compute spectra. This is a very small number, and must mean the authors are doing a large amount of time averaging.

Line 261. Why is this critical? What value can be derived from such a calculation?

Line 269. Maybe instead of "daily" put "diurnal"?

Line 270. The word "differences" is redundant.

Line 271. What is the "inherent daily variability"?

Lines 275-277. Awkward phrasing. Please rewrite!

Lines 280-281. It looks about the same as OSTIA.

Line 306. "larger" This is an apples-to-oranges comparison.

Tables 2 and 3 could be combined with use of parentheses.

Line 349. "...other studies." References needed. Not sure of the point of this sentence.

Line 355. What do the authors mean by "spatial windows"?

Lines 356-357. This is confusing. I think the comparison is going in the opposite direction. That is, for each one-minute saildrone observation, the authors are locating the nearest satellite observation in space and/or time.

Line 363. Repetitive.

Lines 352-365. OK, the three products have somewhat different values of RMSD and correlation (but not bias, why?) for consistently co-located comparisons that for non-consistently. What are we supposed to make of this result? Interpretation needed.

Line 368. I guess this has to be stated, but it seems obvious from the much lower resolution of the SSS products.

Line 380. "0.38degC" Is this a range or a standard deviation?

Line 400. If the CMC product has a Nyquist wavelength of 20 km, the spectrum should not be displayed for scales below that value. That is, much of the green curve in Fig. 7 should not be there. Ditto for the other SST products but with different cutoffs. A similar argument applies to the satellite SSS products. Given that they are sampled on ~25 km grids, the Nyquist cutoff is 50 km, and there is no valid spectral information at scales smaller than that.

Lines 421-422. Repetitive from Table 4. Delete. Ditto lines 411-412.

Lines 435-437. Contradicts lines 426-427. Ditto lines 445-446.

All references should include DOIs if available. This enables the reader to find it.

Lines 578-580. This reference may need a URL. It depends on the editorial policies of RSE.

Author Response

Report 4

Comments and Suggestions for Authors

This paper does a comparison of data from a saildrone mission in spring 2018 to the coast of California and Baja California with satellite SSS and SST measurements. The authors compute RMS differences, bias, signal/noise and correlation. They also compute wavenumber spectra of each measurement. This is a good use for the saildrones, and this kind of comparison makes sense. However, there are a lot of issues with the details of the calculations described in the paper that make it problematic. For reasons described below, the spectra are especially a sticky issue, though more for SSS than SST. The other problem is the description of how the comparisons are carried out. Maybe if they were better described, and I understood better how they were done this could be resolved.

Answer. Thank you to the reviewer for their extremely helpful comments. The paper has been significantly revised to address these issues. We have also added more description on the spectra and redone the figures.

Lines 114-126 contain some problematic plagiarism. I am hoping this is inadvertent on the part of the authors and not repeated in any other parts of the paper.

Answer. We have now corrected the paragraph with proper references. Included are statements concerning the accuracy of Saildrone.

Detailed comments:

Lines 43-44. Technically argo floats are not "buoys". Also, this line repeats on lines 73-74

Answer. Done.

Line 59. What stripes?

Answer. SST infrared data (MODIS) has known stripes due to edge of swath effects, clouds, etc. A reference is given for this for more information.

Line 98. This contradicts line 95, which says SSS is available as an L4 product.

Answer. This was fixed. Level 3 was added to the sentence.

Line 112. "...acoustic Doppler..." Are the ADCP data going to be used in this study? How about the air temperature, etc.? If not, the authors should skip mention of these data to reduce confusion.

Answer. We prefer to leave it in. Although the article does not use ADCP it is still a critical part of the Saildrone instrumentation.  As the primary focus of the paper is to introduce the use of the new Saildrone USV technology for use in conjunction with remote sensing data in coastal regions, we believe it is important to leave in as a precursor to future applications.

Lines 114-126. This text repeats nearly exactly text that appears at the URL given in line 125, so there should be quotation marks with a reference. Even better, delete this entire paragraph, including the URL, and replace with a reference to [21].

Answer. Done. We have now added the URLs to the references.

Lines 132-133. If the measurement has an "inherent spatial resolution" (what does that mean?) of 60 km, what is the justification for gridding at 25 km? Also, when the authors say "All the datasets...", do they mean the three given in the previous paragraph? If so, move this sentence to the previous paragraph.

Answer. A sentence has been added which elaborates on the Saildrone sampling. With respect to the grid resolution, it is common practice for remote sensing data sets to be gridded at different resolutions than their inherent “feature resolution”. In general, the “gridding” resolution is higher than the feature resolution to make sure that the signal feature resolution is not degraded by the gridding.

Line 136. Following this link gives a page with this message at the top:

"Please Note: This dataset is retired and will not be maintained in the future. However, this dataset is displayed for archive purposes and may not reflect the most updated information."

Answer. There was an error as the link pointed to the monthly averages and should have pointed to the 8-day averages that were used in the study. This has been corrected.

The authors should use a more current dataset, and one that will not become unavailable in the  near future. Also, the authors are better off pointing to a DOI in a reference, e.g. reference [21]. (Ditto lines 143, 145, 159, etc.).

Answer. See above comments. The standard practice is to point to the landing pages which contain information on the DOI. URLs have now been moved to the reference section.

Lines 138 and following. This is all very garbled. Again, the authors are taking a 40 km and a 70 km product and gridding it at 25 km (I think they mean 0.25 degrees, which is *not* 25 km). By "The rationale for the two products", do the authors mean the rationale for the 40 km one? Lines 139 and 140 repeat each other. Line 141, smoother than what? The authors need to describe briefly what the real difference is between the 40 and 70 km products.

Answer. A paragraph has been added which now describes in more details the differences between the JPLSMAP and RSS products and the algorithm implementation. As features resolutions are described quoted in kilometers we prefer, for the sake of consistency, to use kilometers.

Line 148. This link is dead. I think PO.DAAC stopped supporting ftp. Since I cannot get to the documents this link is supposed to access, I cannot verify if the ATBD and user guides for the RSS datasets are available at the same location. The authors should do this.

Answer. The link has been updated to reflect the ftp retirement.

Line 172. What is the "forward stream"? Also "10 km".

Answer. Forward stream refers to data produced in the near real time mode from a given date. The 20 km product is not produced anymore, only the 10 km product.

Line 173. So this study only uses the 10 km data?

Answer. Yes.

Section 2.1.3. It's not clear why it is necessary to use so many different SST products. Maybe it will be clearer later. Also, line 150 indicates that three products are being used, whereas I count 6.

Answer. In total statistics were calculated for six products, but figures used only three products as all six products had high correlations and the statistics similar. Only three products were plotted to keep the figures clean, as they were representative of the SST products. We believe it was important to do the comparisons with the six products to show that overall the GHRSST products were comparing very well with Saildrone. This was important to show that the high correlation was not due to one particular product.

Lines 211-225. As indicated from the comments below, this paragraph is not comprehensible and needs to be re-written from scratch.

Answer. This was now rewritten and simplified.

Lines 217-219. Huh? It looks like the different products *are* being evaluated individually.

Answer. This was rewritten.

Line 219. What process?

Answer. The whole section was modified.

Line 220. "calculations were derived"? What does this mean?

Answer. This was referring directly to bias, RMSD, signal to noise rations, and correlations. This has now been highlighted.

Lines 221-225. The abbreviations have already been given, as well as access URLs. Most of this can be deleted. (Ditto lines 226-232)

Answer. Yes, this paragraph was removed because of redundancy.

Line 239. So the criterion was different for the other SST datasets?

Answer. The methodology of the co-location was rewritten. We apologize as this was not at all well explained. A paragraph has now been added which should clarify the co-location. For example, for the SSS data, yes, the co-location depended on the SSS data set used because of the different feature resolutions. Additonally a lot of confusion was caused because the col-location was not clearly elaborated in the text. This is now been fixed.

Line 239. "nearest-neighbor approach" As the saildrones are sampled at one minute intervals (line 120), whereas the satellites are averaged onto one day space-time grid points, this nearest neighbor approach would take the closest one minute saildrone sample (closest in space or in time or in some weighted combination?) to the daily gridded satellite value. This seems like quite a mismatch in time and/or space scales. Perhaps the authors averaged the saildrone data in some way that was not specified in section 2.1.1. I am guessing that each one-minute saildrone sample was compared to a gridded satellite value. This would mean potential large changes in the comparison as the saildrone crossed from one day into another. The details of how this comparison was carried out are very important to the results, so this imprecision needs to be cleared up.

Answer. Yes, we very much apologize for all the confusion this caused. Each one-minute sample was compared with the satellite data set. Unfortunately, there was an error in the Figures 1,2, 5, and 6 were still generated based on the daily averages of the Saildrone data. These figures have now been updated to reflect the co-location with the resolution of Saildrone.

Lines 258-262. Again, the details matter here. Wavenumber spectra? If so, such spectra would normally need to be computed from a linear spatial series or a box. In order to interpret the spectra, we need more information about how they are computed. Almost none is given. For instance, the saildrones are mixing spatial and temporal variability, especially at long wavelengths. For SSS, the satellite is gridded at ~25 km intervals, with a total size of the box being ~500 km north-south. This means 20 samples from which to compute spectra. This is a very small number, and must mean the authors are doing a large amount of time averaging.

Answer. Apologies, yes, we have now added more information on how the spectra were calculated. We have also replotted the spectra to properly reflect the approximate Nyquist wavelength. The spectra were calculated using the assumption of a synoptic scale for the entire Saildrone deployment. Yes, the deployment occurred over two months, a time scale that should not see major changes in the California Current. The main point in the spectra was to show the differences between the SST and SSS, additionally a comparison of the slopes and the consistency with mesoscale variability. We hope to build on this for future studies. This should only be considered an initial step.

Line 261. Why is this critical? What value can be derived from such a calculation?

Answer. Spectral slopes are reflective of the variability of the ocean, as well as cascades from large-scale to small scale or vice versa. Their understanding and comparisons among satellite derived parameters give us a perspective on the ability to resolved the higher resolution spatial (sub-pixel) scales of the oceans.

Line 269. Maybe instead of "daily" put "diurnal"?

Answer. We would prefer to keep “daily”. The rationale is that daily includes both any temporal diurnal variability, but also possible spatial variability resolved by Saildrone on the daily time scale.

Line 270. The word "differences" is redundant.

Answer. First differences removed.

Line 271. What is the "inherent daily variability"?

Answer. The inherent daily variability would have two components, yes, the diurnal variability, but also the spatial variability that Saildrone resolves as it moves one day along its track.

Lines 275-277. Awkward phrasing. Please rewrite!

Answer. Done.

Lines 280-281. It looks about the same as OSTIA.

Answer. These figures have now been replaced with the new figures based on the co-location done at the one-minute sampling of Saildrone. Differences are now more clearly, although visually the SST still are very similar. This is consistent with the high 0.97 correlation.

Line 306. "larger" This is an apples-to-oranges comparison.

Answer. Yes, the salinity products are different. In terms of comparisons with results in the Gulf of Mexico, simply trying to show the consistency in the sign of the biases, most likely associated the different approaches to land contamination.

Tables 2 and 3 could be combined with use of parentheses.

Answer. Table 3 has now been removed as the co-location methodology for SSS fills in all co-location points along the Saildrone track.

Line 349. "...other studies." References needed. Not sure of the point of this sentence.

Answer. Sentences were rewritten

Line 355. What do the authors mean by "spatial windows"?

Answer. This has now been rewritten to explain better the co-location strategy.

Lines 356-357. This is confusing. I think the comparison is going in the opposite direction. That is, for each one-minute saildrone observation, the authors are locating the nearest satellite observation in space and/or time.

Answer. Yes, this section was rewritten. Only Table 2 is included as the stats were run for the co-locations based on the one-minute sampling.

Line 363. Repetitive.

Answer. Yes, this was now rewritten.

Lines 352-365. OK, the three products have somewhat different values of RMSD and correlation (but not bias, why?) for consistently co-located comparisons that for non-consistently. What are we supposed to make of this result? Interpretation needed.

Answer. That is not an accurate statement. The sign of the biases is also different. The differences (now a paragraph has been added to the Methods section describing the differences in the processing of the SSS products) are predominately due to three factors: 1) different approaches to land contamination, 2) smoothness of the data sets, and 3) inherent differences in the algorithms.

Line 368. I guess this has to be stated, but it seems obvious from the much lower resolution of the SSS products.

Answer. Yes. we think it is still important to state up front.

Line 380. "0.38 deg C" Is this a range or a standard deviation?

Answer. RMSD (Root mean square difference)

Line 400. If the CMC product has a Nyquist wavelength of 20 km, the spectrum should not be displayed for scales below that value. That is, much of the green curve in Fig. 7 should not be there. Ditto for the other SST products but with different cutoffs. A similar argument applies to the satellite SSS products. Given that they are sampled on ~25 km grids, the Nyquist cutoff is 50 km, and there is no valid spectral information at scales smaller than that.

Answer. Yes, we have replotted the spectra.  Statements have also been added explaining the Nyquist Wavelengths.

Lines 421-422. Repetitive from Table 4. Delete. Ditto lines 411-412.

Answer. This section has been rewritten. Also made it clear that the resolution of the SSS does not allow for resolving submesoscale variability.

Lines 435-437. Contradicts lines 426-427. Ditto lines 445-446.

Answer. Yes, that was rewritten.

All references should include DOIs if available. This enables the reader to find it.

Answer. Yes, we have checked the references.

Lines 578-580. This reference may need a URL. It depends on the editorial policies of RSE.

Answer. Yes, we have checked the references.

Reviewer 5 Report

This is generally a well written paper describing use of Saildrone measurement off the California and Baja coasts for evaluation of salinity and sea surface temperature products. I have a few comments below which request that additional information be included to more completely describe the work done.

A reference to a more complete description of the Saildrone platform should be included in Section 2.1.1. That section should also include some discussion of the accuracy of the Saildrone measurements of SST and SSS.

I find including the URL for a dataset in the main text distracting while reading the paper and would prefer an appropriate citation instead. PODAAC provides suggested citations for the data products that they host.

Please provide additional information to explain how equation 3 is applied. Is SDROMEAN and mean over a spatial window, a time window or some combination?

The number of collocations for each set of statistics should be given. This is of particular interest when comparing the results of Tables 2 and 3. Are most or all of the RSS70 collocations included in both of those sets while those of JPLSMAP are changed due to differences in coastal coverage?

Are the results for SSS wavelength spectra and coherences derived using the full set of collocations for each satellite and the Saildrone or only those common to all three satellite products?

Author Response

Report 5

Open Review

English language and style

( ) Extensive editing of English language and style required
( ) Moderate English changes required
(x) English language and style are fine/minor spell check required
( ) I don't feel qualified to judge about the English language and style

Yes

Can be improved

Must be improved

Not applicable

Does the introduction provide sufficient background and include all   relevant references?

( )

( )

(x)

( )

Is the research design appropriate?

(x)

( )

( )

( )

Are the methods adequately described?

( )

( )

(x)

( )

Are the results clearly presented?

(x)

( )

( )

( )

Are the conclusions supported by the results?

(x)

( )

( )

( )

Comments and Suggestions for Authors

This is generally a well written paper describing use of Saildrone measurement off the California and Baja coasts for evaluation of salinity and sea surface temperature products. I have a few comments below which request that additional information be included to more completely describe the work done.

Answer. Thank you for your positive feedbacks.

A reference to a more complete description of the Saildrone platform should be included in Section 2.1.1. That section should also include some discussion of the accuracy of the Saildrone measurements of SST and SSS.

Answer. Two references have been added as well as the following paragraph to describe in more detail the accuracies of the Saildrone measurement:

For purposes of validation Saildrone data, during a survey, was compared directly with buoys from the National Data Buoy Center (NDBC). The instrument itself travels at 2 m/s and thus travels 120 m in one minute.

The accuracy of Saildrone USV salinity observations was examined using nearby ship observations 1-10 May 2015 [Cokelet et al., 2015]. The root-mean-square (RMS) salinity difference was found to be 0.01 PSS78. A longer two-month validation was completed in 2018, and all sensors, except the skin SST and barometric pressure where found to be operating within manufacture specifications [Gentemann et al. (2019), submitted].

I find including the URL for a dataset in the main text distracting while reading the paper and would prefer an appropriate citation instead. PODAAC provides suggested citations for the data products that they host.

Answer. References have been added to all the data sets. We have also removed all the URLS. In the acknowledgements we now cite the one URL  http://podaac.jpl.nasa.gov, where all the data and information can be retrieved. Other URLs have been added to the references.

Please provide additional information to explain how equation 3 is applied. Is SDROMEAN and mean over a spatial window, a time window or some combination?

Answer. Sentences have now been added which explain that equation 3 was applied over the time of the deployment and the co-locations.  The equation is applied over the entire Saildrone deployment.

The number of collocations for each set of statistics should be given. This is of particular interest when comparing the results of Tables 2 and 3. Are most or all of the RSS70 collocations included in both of those sets while those of JPLSMAP are changed due to differences in coastal coverage?

Answer. Based on the reviewer’s comments and others we have written, in more detail, the co-location strategy used. We apologize for the confusion, as there was a mistake that figures 1,2,5 and 6 were generated using a nearest neighbor co-location, while the time series (Figures, 3,4) were based on using the full sampling of Saildrone, with each Saildrone sample co-located with satellite pixels. The following was added:

For both SST and SSS satellite products, the co-location with the Saildrone data was conducted by averaging the values of all pixels located within d km from the Saildrone measurement for the same day. d has been selected as the spatial resolution of a given satellite product. Therefore, the spatial co-location window was 1 km for MUR, 5 km for OSTIA and 25 km for JPLSMAP SSS for example. It should be mentioned that co-location in time was restricted to same day measurements regardless of the smoothing used in the Level 3 (SSS) or Level 4 (SST) product. In fact, SMAP SSS products were produced daily but consisted of 8-day running means. These co-location criteria are applied throughout the study; ensuring that more than one pixel is used for the comparison thus reducing any potential impact of noise in the satellite dataset. Co-locations were done for each Saildrone sample to not degrade the resolution of the Saildrone sampling.

Are the results for SSS wavelength spectra and coherences derived using the full set of collocations for each satellite and the Saildrone or only those common to all three satellite products?

Answer. They are based on using all the co-locations. We removed Table 3. The co-location strategy, as mentioned above, allows for matchups to be created for nearly every Saildrone sample.

Round 2

Reviewer 1 Report

The authors have ameliorated their draft in a very elaborated way based on the suggestions of all reviewers.

Author Response

We thank the reviewer for all the time spent and thoughtfullness in reviewing this manuscript. We believe your comments have significantly improved the mansucript.  Thank you!

Reviewer 2 Report

The main points raised by the reviewer have been properly addressed.

A few minor points

Line 160: “The opposite biases…” is already a result of the investigation. Move to appropriate section.

Line 222: delete ‘km’

Line 222: “…centered on the saildrone measurement for the same day”. There were likely several saildrone measurements per day. Were they averaged? Please clarify.

Line 224-225: “… 25 km for JPLSMAP SSS.” Contradicts to lines 367-368. Is it grid resolution or feature resolution? The  latter is 60, 40, 70 km for SMAP JPL, SMAP RSS 40km, SMAP RSS 70km, respectively.  

Lines 364-368 are out of context. Suggest moving to sec 2.2

Author Response

 We thank the reviewer for all the thoroughness and thoughtfullness in reviewing this manuscript. We feel that it has significantly, and in a major way, improved the manuscript.

Open Review

English language and style

( ) Extensive editing of English language and style required
( ) Moderate English changes required
(x) English language and style are fine/minor spell check required
( ) I don't feel qualified to judge about the English language and style

Yes

Can be improved

Must be improved

Not applicable

Does the   introduction provide sufficient background and include all relevant   references?

(x)

( )

( )

( )

Is the research   design appropriate?

(x)

( )

( )

( )

Are the methods   adequately described?

(x)

( )

( )

( )

Are the results   clearly presented?

(x)

( )

( )

( )

Are the conclusions   supported by the results?

(x)

( )

( )

( )

Comments and Suggestions for Authors

The main points raised by the reviewer have been properly addressed.

A few minor points

Line 160: “The opposite biases…” is already a result of the investigation. Move to appropriate section.

Done. Was moved to discussion of Figure 6.

Line 222: delete ‘km’

Done.

Line 222: “…centered on the saildrone measurement for the same day”. There were likely several saildrone measurements per day. Were they averaged? Please clarify.

This sentence has been rewritten to clarify that the pixels were averaged.

Line 224-225: “… 25 km for JPLSMAP SSS.” Contradicts to lines 367-368. Is it grid resolution or feature resolution? The  latter is 60, 40, 70 km for SMAP JPL, SMAP RSS 40km, SMAP RSS 70km, respectively.  

Done. Was corrected.

Lines 364-368 are out of context. Suggest moving to sec 2.2

We feel that lines 364-368 discuss one of the most importance points of the article, i.e. that the differences between the Saildrone derived SST and satellite derived SST can be explained by the inherent daily variability not resolved by the satellites data. However, for SSS, other issues must come into play to explain the differences, including resolution and land contamination. We feel those statements should thus remain in the discussion section and not moved to the Methods section.  

Reviewer 4 Report

This is my second review of this paper. The authors have responded positively to most of my previous review. There are still a lot of problems in the presentation of the manuscript as detailed below.

The authors have provided a link to a retired or obsolete dataset - see line 146. The authors need to go through every link in this paper and make sure it resolves where it is supposed to. The editor should verify that this has been fixed.

*All* references need to include DOIs where available. This item was pointed out in the previous review, but not fixed. The editor should verify that this has been fixed.

*All* references need access dates if appropriate, not "YYYY-MM-DD".

Line 114. Repetitive from line 96.

Line 119. An "acoustic" Doppler current profiler.

Lines 118-120. The authors should add that these data are not being used in this study. Again, I would recommend that this sentence and any reference to the ADCP data just be deleted for brevity and simplicity.

Line 123. "...the Saildrone data *were* compared..."

Lines 123-124. And the results were... Was this comparison done during the Baja campaign that is the subject of this paper? Is there a reference needed?

Lines 124-125. This was already stated in line 113, though with different units. Delete.

Line 127. This for a different deployment in a completely different region, which should be pointed out.

Line 146. Clicking on the link provided in reference [24] goes to a page that has this notice at the top:

"Please Note: This dataset is retired and will not be maintained in the future. However, this dataset is displayed for archive purposes and may not reflect the most updated information."

This issue came up in the previous review. The authors need to be using the most up-to-date versions of these datasets. Are they really using the *monthly* JPLSMAP?

Line 146. This reference also provides access to the dataset itself. Ditto lines 163, 177, etc.

Line 161. Remove sentence starting with "The 70 km..."

Line 164-165. This URL was chopped up during the creation of the pdf.

Line 218. "a spatial km box" Huh?

Line 219. "DELTA-d has been selected as the spatial resolution"? I thought the spatial resolution of a satellite measurement was determined by the satellite itself. Perhaps "has been selected" should be replaced by "is".

Lines 217-228. So each satellite pixel is compared with any one-minute saildrone measurement that happens to occur within the co-location window (line 220) on the same day. Correct? One can see this in the strange jumpiness behavior (see e.g. ~April 20 RSS40 for an extreme example) of the satellite values in Fig. 4. I am guessing those jumps occur when one day turns into another.

Line 255. "differences" is redundant.

Line 268. Technically, since "difference" has a specific mathematical meaning, and SST and SSS are in different units, this does not make sense. Maybe "discrepancies" would be better. Ditto lines 296 and 355.

Line 273. Repetitive from lines 269-270.

Lines 293-295. Any speculation as to why?

Line 301. Reference garbled.

Line 305. I think the authors are referring to actual differences here, not biases as in equation (1). Ditto line 307, the previous paragraph, and lines 313-316.

Line 340. The authors have probably given the wrong reference. I'm not sure what this refers to, but it may be from a completely different region.

Line 345. What is "feature resolution"?

Line 350. This low correlation (0.39) indicates that these products do a very poor job of resolving the variability measured by the saildrone.

Lines 369-370. So basically the time series of Figs. 3 and 4 were converted into spatial series?

Lines 371-371. Not sure what this means. The spectra are representative of the spatial scales of the satellite sampling and greater (lines 373-375).

Line 376. If the Nyquist periods were 120, 80 and 140 km (line 374), then no spectral information is available at scales shorter than this, even if the data are gridded at 25 km. See also lines 446-448.

Line 388 "wavelength"

Line 405. k^-2

Line 415. I'm not sure this is the correct reference, though I have not read this particular paper.

Lines 418-419. This is much more convincing for SST than SSS given the longer Nyquist periods and the low correlation for SSS.

Figs. 9 and 10. The presentation of the error bars in these figures is very awkward. It is not clear why the authors chose to present them for MUR and JPLSMAP and not the other products, or even what they mean - the caption does not say. Almost all of the error bars do not cross zero coherence, which presumably means that the values are all significantly different from zero.

Lines 425-427. I cannot see how the error bars indicate this.

Lines 427-429. This can be seen in the spectra as well. The energy in the saildrone is much larger than the satellite at small scales - even though they have the same slope.

Fig. 9. Coherences should not be displayed for length scales smaller than the Nyquist.

Line 445. Which two spectra?

Author Response

We thank the reviewer for the thoroughness and thoughtfulness of the review. We feel it has significantly improved the manuscript. Thank you.

English language and style

( ) Extensive editing of English language and style required
(x) Moderate English changes required
( ) English language and style are fine/minor spell check required
( ) I don't feel qualified to judge about the English language and style

Yes

Can be improved

Must be improved

Not applicable

Does the   introduction provide sufficient background and include all relevant   references?

(x)

( )

( )

( )

Is the research   design appropriate?

(x)

( )

( )

( )

Are the methods   adequately described?

( )

( )

(x)

( )

Are the results   clearly presented?

( )

( )

(x)

( )

Are the conclusions   supported by the results?

( )

(x)

( )

( )

Comments and Suggestions for Authors

This is my second review of this paper. The authors have responded positively to most of my previous review. There are still a lot of problems in the presentation of the manuscript as detailed below.

The authors have provided a link to a retired or obsolete dataset - see line 146. The authors need to go through every link in this paper and make sure it resolves where it is supposed to. The editor should verify that this has been fixed.

Done. The proper link has now been added to point to V4.2 of the JPLSMAP data. This was the version that we used in the paper. Apologies for the confusion. Thank you very much for finding this mistake.

*All* references need to include DOIs where available. This item was pointed out in the previous review, but not fixed. The editor should verify that this has been fixed.

*All* references need access dates if appropriate, not "YYYY-MM-DD".

Done.  This has now been updated.

Line 114. Repetitive from line 96.

The sentences have now been reworded to avoid the redundancy.

Line 119. An "acoustic" Doppler current profiler.

Done.

Lines 118-120. The authors should add that these data are not being used in this study. Again, I would recommend that this sentence and any reference to the ADCP data just be deleted for brevity and simplicity.

The statement that explicitly says the ADCP data were not used in the study was added. We still would like to leave that sentence in the manuscript as it provides important descriptive information on the Saildrone instrument that might be of interest for future studies.

Line 123. "...the Saildrone data *were* compared..."

Done.

Lines 123-124. And the results were... Was this comparison done during the Baja campaign that is the subject of this paper? Is there a reference needed?

It was now explicitly stated that these comparisons were done during the Baja campaign.

Lines 124-125. This was already stated in line 113, though with different units. Delete.

Done. Sentences were now moved and rephrased to avoid redundancy.

Line 127. This for a different deployment in a completely different region, which should be pointed out.

Done. This is now explicitly stated.

Line 146. Clicking on the link provided in reference [24] goes to a page that has this notice at the top:

"Please Note: This dataset is retired and will not be maintained in the future. However, this dataset is displayed for archive purposes and may not reflect the most updated information."

This issue came up in the previous review. The authors need to be using the most up-to-date versions of these datasets. Are they really using the *monthly* JPLSMAP?

This issue has now been fixed and points to the latest version which was used in the study.

Line 146. This reference also provides access to the dataset itself. Ditto lines 163, 177, etc.

Yes, these references point to the PO.DAAC landing pages for the data sets. The landing pages give information on data access, etc. In addition to the landing pages references on the individual data sets are also included.

Line 161. Remove sentence starting with "The 70 km..."

Done.

Line 164-165. This URL was chopped up during the creation of the pdf.

Hopefully it came out okay now with the conversion to PDF.

Line 218. "a spatial km box" Huh?

Done. The sentence has been rephrased.

Line 219. "DELTA-d has been selected as the spatial resolution"? I thought the spatial resolution of a satellite measurement was determined by the satellite itself. Perhaps "has been selected" should be replaced by "is".

Although Level 3 and 4 data sets are gridded at the same resolution they each have different “feature” resolutions. Feature resolution is defined as the actual resolution the data set can resolve with respect to the ocean spatial variability.  This is especially important for the SSS data sets. Although they are all gridded at 25km, they each have different “feature” resolutions due to differences in processing, including application of smoothing to reduce noise.

Lines 217-228. So, each satellite pixel is compared with any one-minute saildrone measurement that happens to occur within the co-location window (line 220) on the same day. Correct? One can see this in the strange jumpiness behavior (see e.g. ~April 20 RSS40 for an extreme example) of the satellite values in Fig. 4. I am guessing those jumps occur when one day turns into another.

That is essentially correct. But it is important too (was added to the current version of the manuscript) to clarify that each of the co-located points are not necessarily independent. This is because of the one-minute Saildrone sampling and, thus, possible overlapping satellite pixels that are used to calculate the matchup.

Line 255. "differences" is redundant.

We believe the following is the paragraph that the reviewer was referring to:

“A major question to be answered was how the observed differences between the satellite-derived and Saildrone products are related to unresolved subpixel scale spatial variability. To further examine the differences between the satellite-derived products and Saildrone measurements, and their relationship to spatial scales, the wavelength spectra were examined for consistency with known spectral slopes defining mesoscale-submesoscale variability. Overall, the goal was to connect these differences to possible issues of spatial resolution inherent to each dataset. “Although the word “difference” is used twice we feel it is necessary to explain the main point of the paper, which is introducing the Saildrone technology for validation of satellite derived products.

Line 268. Technically, since "difference" has a specific mathematical meaning, and SST and SSS are in different units, this does not make sense. Maybe "discrepancies" would be better. Ditto lines 296 and 355.

Done. The word was changed to discrepancy.

Line 273. Repetitive from lines 269-270.

We feel this is not completely repetitive. The previous lines introduce the subject of examining the differences (discrepancies) while these lines start examining the results.

Lines 293-295. Any speculation as to why?

The following sentence was added: “ Understanding these biases is beyond the scope of this work. Possible explanations for such biases would include air-sea interactions.  Other issues could include biases in input data as different sensors are used in the optimal interpolation”. Overall an analysis of the actual differences and their causes should be the focal point of future work.

Line 301. Reference garbled.

Done.

Line 305. I think the authors are referring to actual differences here, not biases as in equation (1). Ditto line 307, the previous paragraph, and lines 313-316.Yes, that is correct, but we would like to continue to use the word bias in 313 to 316 as it reflects an important conclusion from figures 5,6, where along the Saildrone Track one sees warm, cold, fresh, and salty biases.  The difference is that equation 1) reflects the mean bias over the entire track, as is stated explicitly in the text.

Line 340. The authors have probably given the wrong reference. I'm not sure what this refers to, but it may be from a completely different region.

Fixed.

Line 345. What is "feature resolution"?

Feature resolution is different from  grid resolution. Feature resolution refers to actual spatial scales that a given  product can resolve. These are usually not the same due to smoothing in space and time required to generate gap free data.In the case of SSS it is very important to distinguish the difference between grid and feature resolution. All three products are gridded at 25 km but their feature resolution is different. Thus, the terminology used RSS40 and RSS70.  In this case the feature resolution for RSS40 is 40 kilometers and RSS70 70 kilometers.  For JPLSMAP the feature resolution is close to 60 km. 

Line 350. This low correlation (0.39) indicates that these products do a very poor job of resolving the variability measured by the saildrone.

Yes, the words very poor are not used, but it is made clear that there are two reasons for this:

1)    The issue of land contamination.

2)    The  spatial resolution is not adequate for resolving the mesoscale and submesoscale variability associated with the region.

Lines 369-370. So basically the time series of Figs. 3 and 4 were converted into spatial series?

Yes. This was done to visualize better where the differences between the Saildrone and satellite derived products where occurring with respect to distance from the coast along the Saildrone track.

Lines 371-371. Not sure what this means. The spectra are representative of the spatial scales of the satellite sampling and greater (lines 373-375).

The exact sentence is: “the spectra are representative of the approximate sampling of the Saildrone 1-minute samples”. The spectra where calculated for the resolution of the Saildrone samples. The satellite spectra were also calculated based on the co-locations with the  full resolution Saildrone sampling.

Line 376. If the Nyquist periods were 120, 80 and 140 km (line 374), then no spectral information is available at scales shorter than this, even if the data are gridded at 25 km. See also lines 446-448.Yes, that is correct. For the sake of plotting, based on the 25 km grid spacing, we still showed the spectra to 50 km. The coherence for SSS is very clear and consistent in showing there is not coherence at scales <100km.

Line 388 "wavelength"

Done.

Line 405. k^-2Done.

Line 415. I'm not sure this is the correct reference, though I have not read this particular paper.

Done. Changed to [41].

Lines 418-419. This is much more convincing for SST than SSS given the longer Nyquist periods and the low correlation for SSS.

The following sentence was added to clarify the statement:
However, the SSS is limited to scales greater than 100km. This is further confirmed by the coherences.

Figs. 9 and 10. The presentation of the error bars in these figures is very awkward. It is not clear why the authors chose to present them for MUR and JPLSMAP and not the other products, or even what they mean - the caption does not say. Almost all of the error bars do not cross zero coherence, which presumably means that the values are all significantly different from zero.

We chose to only show the error bars for two of the products to be representative of the statistical significance. Showing all the error bars would be too messy. The main point of the errors was to show that for scales < 100 km the SSS spectra essentially become statistically insignificant, consistent with the spectra and of course the Nyquist wavelengths.  Overall the coherences do become more uncertain at the smaller wavelengths. Error bars do cross the zero threshold at lower wavelengths.

Lines 425-427. I cannot see how the error bars indicate this.

Both the SST and SSS spectra show peaks at 100 km with error bars indicating the statistical significance. The SST spectra also has a peak at 300 km which is statistically significant. In all cases the error bars are bar above the 0 threshold.

Lines 427-429. This can be seen in the spectra as well. The energy in the saildrone is much larger than the satellite at small scales - even though they have the same slope.

Yes, also indicative that Saildrone is resolving scales not seen either by the SST or SSS data.

Fig. 9. Coherences should not be displayed for length scales smaller than the Nyquist.

The only SST data with a Nyquist larger than 10 km would be the CMC data set (20 km). The error bars and coherences at these scales clearly indicate that there is little to no relationship between the Saildrone SST and satellite products at these scales.  For the SST spectra we did cutoff CMC at 20 km.

Line 445. Which two spectra?

This was now rewritten to indicate “the major difference between the SST and SSS spectra”.

Round 3

Reviewer 4 Report

This is my third review of this paper. I am giving it rating of "minor revision" because I do not want to review it again. However, there are still a lot of problems with it as indicated by the volume of comments below.

The authors have added DOIs as I have requested (and RS editorial policies probably dictate). I tried a couple of them at random and found they were not correct. The authors need to go through the entire reference list to make sure these numbers are accurate as they help others quickly find the references.

Line 117. And the results of those comparisons were... I asked for this to be described in my previous review. The authors have refused to elaborate, but do not say why. The authors should show the locations of the buoys on the maps of Fig. 1, say.

Lines 124-125. Replace with "SSS and SST data are used in this study, but the ADCP and other environmental data are not."

Line 131. "...were found..."

Lines 135-136. Again, I would delete this. OK, the authors feel the need to mention the presence of ADCP data in the data stream which they do above, but they do not need to give this kind of detail. It's distracting.

Line 160. Repetitive from lines 144-145.

Lines 217-218. "DELTA-d is defined as..."? I do not see that this variable used later on in the paper, so I'm not sure of the point of this sentence.

Lines 218-219. Give the spatial co-location windows for all the satellites.

Line 254. The word "differences" is redundant. The word "difference" is the final "D" in RMSD. The authors seem to have mis-interpreted my previous comment. Just delete this word.

Line 267. Again, I would use the word "discrepancies" here. Ditto line 299. And line 360. I asked the authors to change this in the previous review. I do not know why they did not do so.

Line 269. Fresher than what? There is a comparison being done here, but it's not clear between what and what.

Lines 269-271. The saildrone SSS actually looks quite different from the satellite products - to my eye. Anyway, this sentence contradicts itself.

Line 372. Does "sub-daily" refer to lower frequency than daily, or shorter time scale than daily? Either way the next sentence does not follow as diurnal variability is daily, not sub-daily whatever that means.

Line 376. The first 500 km of the mission? Where on the Baja coast does this cut off? Maybe around 33N?

Line 380. Take out "For the sake of," In fact this sentence could be removed as it is obvious from the figure.

Line 402. Table 3? This sentence is repetitive from line 400.

Line 407. And from the fact that only 500 km of saildrone record were used (line 376).

Line 422. The SSS wavelength spectra?

Line 425. "greater than 100 km due to the low resolution of the satellites."

Line 429. Can the authors speculate about why there is a minimum around 120 km?

Figs. 9 and 10. The basis for the error bars is not explained. Why error bars on the coherences, but not on the spectra? Why just have error bars on the MUR/JPLSMAP product and not the others? (This latter is explained in the rebuttal, but not in the figure caption.)

Line 433. They seem significant, though not large below 30 km. That is, the error bars do not reach the zero line.

Line 444. Larger than the SSS coherences? I do not know what this sentence is trying to say.

Lines 461-462. Contradicts line 433.

Lines 460-461. Repetitive from lines 379-380.

Line 464. The SSS satellites do not resolve this at all - and are not designed to.

Line 470. "...a positive bias...". "...RMSDs are..."

Line 479. "...as part...is..."

Line 497. Significant differences between each other? With the saildrone?

Line 531. DOI does not work.

Line 565. DOI does not work.

Author Response

We thank the reviewer for the through review. Below is the response to the reviewer's comments:

Journal

Remote Sensing (ISSN 2072-4292)

Manuscript ID

remotesensing-517332

Type

Article

Number of Pages

37

Title

Using Saildrones to Validate Satellite-Derived Sea Surface Salinity and Sea Surface Temperature along the California/Baja Coast

Authors

Jorge Vazquez-Cuervo * , Jose Gomez-Valdes , Marouan Bouali , Luis Miranda , Tom Van der Stocken , Wenquing Tang , Chelle Gentemann

Abstract

Traditional ways of validating satellite-derived sea surface temperature (SST) and sea surface salinity (SSS) products, using comparisons with buoy measurements, do not allow for evaluating the impact of mesoscale to submesoscale variability. Here we present the validation of remotely-sensed SST and SSS data against the unmanned surface vehicle (USV) – Saildrone – measurements from the Spring 2018 Baja deployment. More specifically, biases and root mean square differences (RMSD) were calculated between USV-derived SST and SSS values, and six satellite-derived SST (MUR, OSTIA, CMC, K10, REMSS, and DMI) and three SSS (JPLSMAP, RSS40, RSS70) products. Biases between the USV SST and OSTIA/CMC/DMI were approximately zero while MUR showed a bias of 0.2°C. OSTIA showed the smallest RMSD of 0.36°C while DMI had the largest RMSD of 0.5°C. An RMSD of 0.4°C between Saildrone SST and the satellite-derived products could be explained by the daily variability in USV SST which currently cannot be resolved by remote sensing measurements. For SSS, values from the JPLSMAP product showed saltier biases of 0.2 PSU, while RSS40 and RSS70 showed fresh biases of 0.3 PSU. An RMSD of 0.4 PSU could not be explained solely by the daily variability of the USV-derived SSS. Coherences were significant at the longer wavelengths, with a local maximum at 100 km that is most likely associated with the mesoscale turbulence in the California Current System.

Bottom of Form

Top of Form

Author's Reply to the Review Report (Reviewer 4)

Please provide a point-by-point response to the reviewer’s comments and either enter it in the box below or upload it as a Word/PDF file. Please write down "Please see the attachment." in the box if you only upload an attachment. An example can be found here.

* Author's Notes to Reviewer

Bottom of Form

0 words

Word / PDF

or

Top of Form

Review Report Form

Open Review

English language and style

( ) Extensive editing of English language and style required
( ) Moderate English changes required
(x) English language and style are fine/minor spell check required
( ) I don't feel qualified to judge about the English language and style

Yes

Can be improved

Must be improved

Not applicable

Does the introduction provide sufficient background and include all relevant references?

(x)

( )

( )

( )

Is the research design appropriate?

(x)

( )

( )

( )

Are the methods adequately described?

( )

(x)

( )

( )

Are the results clearly presented?

( )

(x)

( )

( )

Are the conclusions supported by the results?

(x)

( )

( )

( )

Comments and Suggestions for Authors

This is my third review of this paper. I am giving it rating of "minor revision" because I do not want to review it again. However, there are still a lot of problems with it as indicated by the volume of comments below.

The authors have added DOIs as I have requested (and RS editorial policies probably dictate). I tried a couple of them at random and found they were not correct. The authors need to go through the entire reference list to make sure these numbers are accurate as they help others quickly find the references.

Line 117. And the results of those comparisons were... I asked for this to be described in my previous review. The authors have refused to elaborate, but do not say why. The authors should show the locations of the buoys on the maps of Fig. 1, say.

The reviewer asked originally about the accuracy of Saildrone instryment. This is answered in lines 126-131. This instruments for the Baja cruise were the same, and thus the accurach would be the same. The point about mentioning the buoys was simply to respond to the reviewer that independent validation to confirm that all instruments were working for the Baja Deployment. As the buoys were not a part of this study (references on the validation of Saildrone were added earlier based on the reviewers recommendation) it would be confusing to add them to Figure 1. We see no value.

Lines 124-125. Replace with "SSS and SST data are used in this study, but the ADCP and other environmental data are not."

Done.

Line 131. "...were found..."

Done.

Lines 135-136. Again, I would delete this. OK, the authors feel the need to mention the presence of ADCP data in the data stream which they do above, but they do not need to give this kind of detail. It's distracting.

Statement has been removed. It is normal data center policy to give full descriptions of the data sets.  The statement has been removed to avoid confusion.

Line 160. Repetitive from lines 144-145.

Statement 160 was removed to avoid redundancy.

Lines 217-218. "DELTA-d is defined as..."? I do not see that this variable used later on in the paper, so I'm not sure of the point of this sentence.

Delta D was added to the previous sentence to clarify the definition and why it is used. We believe just saying delta-d adds clarity to the how the spatial co-location was done.

Lines 218-219. Give the spatial co-location windows for all the satellites.

Done.

Line 254. The word "differences" is redundant. The word "difference" is the final "D" in RMSD. The authors seem to have mis-interpreted my previous comment. Just delete this word.

Done.

Line 267. Again, I would use the word "discrepancies" here. Ditto line 299. And line 360. I asked the authors to change this in the previous review. I do not know why they did not do so.

Done.

Line 269. Fresher than what? There is a comparison being done here, but it's not clear between what and what.

All comparisons are done between the satellite derived products and Saildrone.  Sentence was reworded that hopefully clarifies this. Fresher means that the satellite derived product is less salty than Saildrone.

Lines 269-271. The saildrone SSS actually looks quite different from the satellite products - to my eye. Anyway, this sentence contradicts itself.

We do not understand were the contradiction lies as this is clearly stated in the following sentence:

Even before statistics are calculated between the products, one can visually observe a strong similarity between the Saildrone CTD SST and the satellite SST products, while the same imagery of the SSS indicates that pronounced biases exist between the products. Overall, one identifies visually the fresh bias between the RSS40, RSS70 and the Saildrone SSS

Line 372. Does "sub-daily" refer to lower frequency than daily, or shorter time scale than daily? Either way the next sentence does not follow as diurnal variability is daily, not sub-daily whatever that means.

To avoid confusion we added a statement that defines sub-daily as high frequencies than than the diurnal variability which Saildrone would be able to resolve.  The point we wanted to cmphasize is that Saildrone is, in fact, also resolving sub-daily time scales that neither of the satellite derived products can resolve.

Line 376. The first 500 km of the mission? Where on the Baja coast does this cut off? Maybe around 33N?

Yes that would be the approximate cutoff. We used the Southward leg also as it was the time period when Saildrone was closest to the coast and capturing the upwelling variability.

Line 380. Take out "For the sake of," In fact this sentence could be removed as it is obvious from the figure.

The sentence was removed.

Line 402. Table 3? This sentence is repetitive from line 400.

There is no mention of table 3 in line 402. It is referred to once in line 409. Not sure what the reviewer is referring to.

Line 407. And fromj the fact that only 500 km of saildrone record were used (line 376).

Line 422. The SSS wavelength spectra?

Yes it is clearly stated that it is for wavelengths, thus wavelength spectra.

Line 425. "greater than 100 km due to the low resolution of the satellites."

This statement is not a line 425. 200km was changed to 100km.

Line 429. Can the authors speculate about why there is a minimum around 120 km?

It would be hard to speculate and prefer not add to the document. One possibility is the boundary between wavelengths associated with the variability of California Current system and the msesoscale variability associated with  fronts and  eddies.  We hope to build on this for the future.

Figs. 9 and 10. The basis for the error bars is not explained. Why error bars on the coherences, but not on the spectra? Why just have error bars on the MUR/JPLSMAP product and not the others? (This latter is explained in the rebuttal, but not in the figure caption.)

We felt that adding error bars to the spectra was not necessary, as the primary point of the spectra was to show the consistency of the slope. However, error bars for the coherence are necessary to determine the statistical significance.

Line 433. They seem significant, though not large below 30 km. That is, the error bars do not reach the zero line.

The statistical significance is very weak at best. Thus the general statement about lack of coherence at the shorter wavelengths we feel is still correct.

Line 444. Larger than the SSS coherences? I do not know what this sentence is trying to say.

Here is the statement on Line 444. No reference to SSS coherences. Not sure what the reviewer is referring to: : “Again, this indicates that the satellite-derived products are not fully resolving the spatial scales associated with submesoscale variability. Overall the SST coherences are larger for scales of less than 100 km, but the error bars indicate that appropriate caution be taken in interpreting the significance of these findings.”

This is simply referring for scales greater than 100km the SST coherences are clearly higher, greater than 0,8.  For Salinity no such high coherences exist for scales longer than 100km. A possible explanation is given for this that issues of land contamination in the SSS are preventing a full resolution of the California Current system.

Lines 461-462. Contradicts line 433.

They are consistent. The spectra for SSS is cutoff at 50km because all the SSS products are gridded at 25km.  The fact that there is really no statistically significant coherence in the SSS products for wavelengths less than 100km is completely consistent with the Nyquist wavelengths for the SSS products.

Lines 460-461. Repetitive from lines 379-380.

460-461 has been deleted.

Line 464. The SSS satellites do not resolve this at all - and are not designed to.

Yes that is correct, and this research presents further validation of that statement. However, this paper can also be used as a reference as new SSS products with better land contamination and improved resolution could improve the quality of SSS in coastal regions.

Line 470. "...a positive bias...". "...RMSDs are..."

Positive biases were changed to “warm biases”. RMSDs of 0.4 to 0.5 was added.

Line 479. "...as part...is..."

Changed to: . Land contamination results when part of the satellite footprint is over land

Line 497. Significant differences between each other? With the saildrone?

Saildrone was added.

Line 531. DOI does not work.

All the DOI have now been checked.

I tested the doi and it did work. Believe the problem might be that the

Abbreviation for doi: is used one needs to use: http://doi.org/ to access the proper DOI.

Line 565. DOI does not work.

All the DOI have now been checked.
